# SANIA$^*$: Polyak-type Optimization Framework Leads to Scale Invariant Stochastic Algorithms

## Abstract

Adaptive optimization methods are widely recognized as among the most popular approaches for training Deep Neural Networks (DNNs). Techniques such as Adam, AdaGrad, and AdaHessian utilize a preconditioner that modifies the search direction by incorporating information about the curvature of the objective function. However, despite their adaptive characteristics, these methods still require manual fine-tuning of the step-size. This, in turn, impacts the time required to solve a particular problem. This paper presents an optimization framework named **SANIA** to tackle these challenges. Beyond eliminating the need for manual step-size hyperparameter settings, SANIA incorporates techniques to address poorly scaled or ill-conditioned problems. We also explore several preconditioning methods, including *Hutchinson's method*, which approximates the Hessian diagonal of the loss function. We conclude with an extensive empirical examination of the proposed techniques across classification tasks, covering both convex and non-convex contexts.

## 1 Introduction

Machine Learning (ML), especially Deep Neural Networks (DNNs), has emerged as a transformative tool, setting the stage for unprecedented advances across many disciplines, including computer vision Krizhevsky et al. (2012); Simonyan & Zisserman (2014); He et al. (2016) and natural language processing Wolf et al. (2020); Mikolov et al. (2013); Devlin et al. (2018); Radford et al. (2018), as well as science Xie & Grossman (2018); Gómez-Bombarelli et al. (2018); Kaliyev et al. and engineering Bello et al. (2016); LeCun et al. (1990); Eshkevari et al. (2021); Gulgec et al. (2020; 2017) to name a few.

The enormous potential of these models is enabled through the efficacy of the optimization methods that train them. In the domain of ML the training task can be expressed as solving the following problem

$$\min_{w \in \mathbb{R}^d} f(w) := \frac{1}{n} \sum_{i=1}^n f_i(w), \tag{1}$$

where $w \in \mathbb{R}^d$ represents the weight parameter, and each $f_i : \mathbb{R}^d \to \mathbb{R}$ is a sufficiently smooth function. To provide a practical context, consider a dataset denoted as $\{(x_i, y_i)\}_{i=1}^n$, where $x_i \in \mathbb{R}^d$ is the data sample and $y_i \in \mathbb{R}$ represents the label corresponding to that sample. If $f_i(w) = \frac{1}{2}(x_i^T w - y_i)^2$, this optimization problem gives rise to the well-known least squares problem. Similarly, if $f_i(w) = \log(1 + e^{-y_i x_i^T w})$, we get logistic regression problem.

**Stochastic Gradient Descent.** To address problem equation 1, one of the fundamental techniques employed is Stochastic Gradient Descent ($SGD$) Robbins & Monro (1951); Polyak (1990); Polyak & Juditsky (1992); Nemirovski et al. (2009); Bottou et al. (2018). This method iteratively updates the weight parameter $w$ according to the following scheme:

$$w_{t+1} = w_t - \gamma_t \nabla f_i(w_t), \tag{2}$$

---

$^*$SANIA is an abbreviation formed from letters of working title of this paper: **ScAliNg Invariant Algorithm.**

where $\gamma_t$ is the step-size schedule and $i \subset [n] := \{1, 2, \dots, n\}$ is chosen uniformly as random. Unfortunately, the optimal step-size[1] schedule often relies on problem-specific parameters, such as the Lipschitz-smoothness constant and the level of stochastic gradient noise, which are frequently not accessible. Consequently, achieving an optimal step-size typically demands a substantial amount of tuning, which can be quite costly in practical applications. Numerous methodologies have been developed to tackle this issue. One of the first approaches that reduces the number of parameters to tune is the AdaGrad method by Duchi et al. (2011); Li & Orabona (2019); Ward et al. (2020). An additional challenge arises from the fact that using the same learning rate for each feature $j \in [d]$ might not yield the best performance. To address this, diagonal preconditioning techniques have been employed in the SGD setting by methods such as AdaGrad by Duchi et al. (2011), RMSProp by Tieleman et al. (2012), Adam by Kingma & Ba (2015), AMSGrad by Reddi et al. (2018), AdamW by Loshchilov & Hutter (2019), AdaHessian by Yao et al. (2021), AdaDelta by Zeiler (2012), and OASIS by Jahani et al. (2022). However, all of these methods still require a considerable degree of parameter tuning to achieve optimal performance. Another approach is associated with parameter-free regret minimization for online learning problems, as discussed in various papers Mcmahan & Streeter (2012); McMahan & Orabona (2014); Orabona & Pál (2016); Orabona & Tommasi (2017); Orabona (2019); Carmon & Hinder (2022); Ivgi et al. (2023); Defazio & Mishchenko (2023); Cutkosky et al. (2023); Mishchenko & Defazio (2023). Finally, in our paper, we explore the Stochastic Polyak step-size approach as an adaptive parameter-free method.

**Stochastic Polyak step-size (SPS) Methods.** Polyak step-size method was first proposed by Polyak (1969; 1987) for non-smooth problems. Recently, stochastic Polyak step-size was proposed by Oberman & Prazeres (2019); Berrada et al. (2020); Loizou et al. (2021); Gower et al. (2021); Orvieto et al. (2022). Subsequently, lots of variants of SPS have emerged, such as mSPS by D'Orazio et al. (2021) and AdaSLS by Jiang & Stich (2023). To further relax the requirements for interpolation condition in SPS, many attempts have been made by Gower et al. (2022); Orvieto et al. (2022); Garrigos et al. (2023); Schaipp et al. (2023). A variant of second-order expansion for SPS was presented by Li et al. (2023). Next we describe the main idea of Polyak step-size in more detail.

To derive the deterministic Polyak step-size, let us consider a convex function $f(w)$ and the step equation 2. We obtain the step-size from the following upper-bound on the distance from the current point $w_{t+1}$ to the minimum $w^*$:

$$\|w_{t+1} - w^*\|^2 = \|w_t - w^*\|^2 + \|\gamma_t \nabla f(w_t)\|^2 - 2\gamma_t \langle \nabla f(w_t), w_t - w^* \rangle$$
$$\leq \|w_t - w^*\|^2 + \gamma_t^2 \|\nabla f(w_t)\|^2 - 2\gamma_t(f(w_t) - f(w^*)).$$

Minimizing the right hand side by $\gamma_t$, we get: $\gamma_t = \frac{f(w_t) - f(w^*)}{\|\nabla f(w_t)\|^2}$. Similarly, in the stochastic case, the Stochastic Polyak step-size (SPS) is defined as

$$w_{t+1} = w_t - \frac{f_i(w_t) - f_i^*}{\|\nabla f_i(w_t)\|^2} \nabla f_i(w_t), \tag{3}$$

where $f_i^*$ is a minimal value of function $f_i(w)$. Another way to derive this formulation is by solving the following optimization problem:

$$w_{t+1} = \underset{w \in \mathbb{R}^d}{\arg\min} \|w - w_t\|_2^2, \text{ s.t. } f_i(w_t) + \langle \nabla f_i(w_t), w - w_t \rangle = f_i^*, \tag{4}$$

where equation 3 is an explicit formulation of equation 4. In case where the value of $f_i^*$ is known and set to 0 for all $i$ (a common scenario in over-parameterized deep neural networks) we obtain this simplified expression for equation 3: $w_{t+1} = w_t - \frac{f_i(w_t)}{\|\nabla f_i(w_t)\|^2} \nabla f_i(w_t)$. This condition, referred to as the "interpolation condition", is expressed as $f_i^* = 0$.

**Preconditioning / Feature scaling.** Preconditioning is a technique used to improve the convergence rate of algorithms applied to data that may exhibit poor scaling or ill-conditioning. Algorithms leveraging preconditioning typically follow a generic update rule, which can be expressed as

$$w_{t+1} = w_t - \gamma_t B_t^{-1} m_t, \tag{5}$$

---

[1]In this work, we focus on the minimization of empirical loss equation 1. We refer to recent studies that discuss how the step size can influence the generalization error Kaur et al. (2023); Wu & Su (2023); Ma & Fattahi (2022); Chen & Bruna (2023).

where $B_t \in \mathbb{R}^{d \times d}$ is an invertible positive definite matrix, and $m_t$ is a gradient or its approximation. The origin of such a step is Newton method by Newton (1687); Raphson (1697); Kantorovich (1948a;b; 1949) which uses the exact Hessian to precondition the gradient of the objective function, i.e. $B_t = \nabla^2 f(w_t)$ and $m_t = \nabla f_i(w_t)$. Newton method can be very effective for minimizing convex objectives. However, the prohibitive cost of computing and inverting the Hessian matrix, together with issues around negative eigenvalues, makes this approach impractical for machine learning tasks. To address this issue, one can use methods that never define the Hessian of the objective function explicitly but rather use its approximation or solve the Newton system using iterative algorithms (Martens et al., 2010).

**Quasi-Newton methods (QN).** Methods that construct an approximation of the (inverse) Hessian date back to the 70s such as BFGS (Broyden, 1967; Fletcher, 1970; Goldfarb, 1970; Shanno, 1970), L-BFGS (Nocedal, 1980; Liu & Nocedal, 1989), and SR-1 (Conn et al., 1991; Khalfan et al., 1993). These optimization methods take advantage of a cheap way to build (inverse) Hessian matrix estimation algorithms based on past gradient information. One of the most prominent QN method is *Symmetric Rank 1(SR-1)* which recursively approximates the Hessian as follows: $B_{t+1} = B_t + \frac{(y_t - B_t s_t)(y_t - B_t s_t)^\top}{(y_t - B_t s_t)^\top s_t}$, where $s_t = w_{t+1} - w_t$ and $y_t = \nabla f_i(w_{t+1}) - \nabla f_i(w_t)$. Although, SR-1 update only makes a rank-1 change to the previous Hessian approximation and evidently has a simple form, in practice it displays better convergence to the true Hessian than other similar methods like BFGS (Nocedal & Wright, 2006, p.145). Another useful property of this approximation is self-complementarity, which means that we can find the inverse Hessian approximation $B_t^{-1}$ using the same vector pair $s_t$ and $y_t$: $B_{t+1}^{-1} = B_t^{-1} + \frac{(s_t - B_t^{-1} y_t)(s_t - B_t^{-1} y_t)^\top}{(s_t - B_t^{-1} y_t)^\top y_t}$. Note, that this approximation method does *not* necessarily generate a positive definite matrix.

**Contributions.** Before delving into the details, we outline the primary contributions of this work:

- We present the General Framework for Preconditioned and Second-order Polyak methods. This framework covers classical optimization methods, provides valuable insights into Polyak step-size methods, and enables the development of novel Polyak step-size methods.

- We propose the first Stochastic Cubic Newton method with Polyak step-size.

- We introduce the new scale invariant versions of AdaGrad and Adam, which make them invariant to some basis transformations.

- We conduct comprehensive experiments encompassing a diverse range of scenarios, including both convex and non-convex settings.

**Organisation.** In this paper, we have consolidated our findings and integrated them into a comprehensive framework presented in Section 2. Additionally, Section 3 offers a detailed presentation of the results from our experiments.

**Notation and Assumptions.** We introduce the notation used throughout the paper and state the underlying assumptions that guide our analysis. We equip the primal space $w \in \mathbf{E}$ and the dual space $g \in \mathbf{E}^*$ with the conjugate norms $\|w\|$ and $\|g\|_*$, respectively. As a special case, for a positive definite matrix $B \in \mathbb{R}^{d \times d}$, we introduce the conjugate Euclidean norms as follows: $\|w\|_B = \langle Bw, w \rangle^{1/2}$ and $\|g\|_{B^{-1}} = \langle g, B^{-1}g \rangle^{1/2}$. As an example, $\nabla f(w) \in \mathbf{E}^*$ and $\nabla^2 f(w)h \in \mathbf{E}^*$ for $h \in \mathbf{E}$. We define the operator $\odot$ as a component-wise product between two vectors, also known as the Hadamard product. For the vector $w$, $w^2$ and $\sqrt{w}$ means component-wise square and square root, respectively. We represent diag$(w)$ as a diagonal matrix of a given vector $v$ and a vector diagonal$(H) \in \mathbb{R}^d$ as the diagonal of a matrix $H \in \mathbb{R}^{d \times d}$. For simplicity, we denote $g_t = \nabla f_i(w_t)$ and $H_t = \nabla^2 f_i(w_t)$ if it is not defined differently. Also, we denote an action of the linear operator as $B[h]^2 = \langle Bh, h \rangle$.

**Interpolation Condition.** The *Interpolation Condition* is an assumption often applied in optimization and machine learning, particularly in the analysis of overparameterized models such as deep neural networks. It assumes the existence of a set of model parameters $w^*$ such that the loss function $f(w)$ achieves its infimum across all data points. This condition is indicative of a scenario where the model has sufficient flexibility to perfectly fit the training data, leading to zero loss for every data point. Such regimes are

commonly encountered in overparameterized deep neural networks Ma et al. (2018b); Zhang et al. (2021) or non-parametric regression models Liang & Rakhlin (2020); Belkin et al. (2019), where the model's capacity exceeds the complexity of the data, ensuring exact interpolation of the training set. This is one of the standard assumptions in analysis of methods with the Stochastic Polyak step-size e.g. Schaipp et al. (2023); Loizou et al. (2021); Gower et al. (2022); Li et al. (2023); Orvieto et al. (2022). Unless otherwise stated, our default assumption is that Assumption 1 holds true.

> **Assumption 1: Interpolation Condition**
>
> We assume that the *interpolation* condition holds for a set of non-negative functions $\{f_i(w)\}_{i=1}^n$ ($f_i(w) \geq 0 \ \forall w \in \mathbf{E}$), when $\exists w^* \in \mathbf{E}$, s.t. $f(w^*) = 0$. Consequently, $f_i(w^*) = 0$ for all $i = 1, 2, ..., n$.

## 2 SANIA – general framework

### 2.1 General framework

In this section, we propose a general framework equation 6 for preconditioning stochastic Polyak step-size methods. This framework generalizes some well-known first-order, second-order, and Quasi-Newton methods from Polyak step-size perspective. The main feature of the framework is that it highlights some insights about SPS and provides an instrument to generalize existing methods as Polyak step-size methods. It makes them adaptive and parameter-free in the SPS setting. The generality of this framework makes it difficult to propose an explicit step. Therefore, we will focus on the most promising cases and provide their explicit formulations to introduce new methods. In the following section we will demonstrate the problem settings required to derive existing and proposed methods using SANIA equation 6. We note that if any particular variable from the General Framework is not mentioned explicitly it is assumed to be fixed at zero.

> **Definition 1: SANIA: General Framework**
>
> Let $B_t \succ 0$ and $D_t$ be symmetric matrices, and $\tau_t$ be sequence of numbers that is given or can be computed for any given $t \geq 0$. We consider the following minimization problem:
>
> $$w_{t+1}, \alpha_{t+1} = \underset{w \in \mathbb{R}^d, \alpha \in \mathbb{R}}{\arg\min} \frac{1}{2}\|w - w_t\|_{B_t}^2 + \tau_t \alpha$$
>
> $$\text{s.t.} \quad f_i(w_t) + \langle m_t, w - w_t \rangle + \frac{1}{2}\langle D_t(w - w_t), w - w_t \rangle \leq \alpha. \tag{6}$$
>
> Note that $B_t$ is required be a positive definite matrix to ensure that $\|\cdot\|_{B_t}$ is a Euclidean norm.

### 2.2 Existing methods

**SGD.** Let us first derive an update rule for the most frequently used variant of Stochastic Gradient Descent (SGD) method using SANIA equation 6.

We set parameters as follows:

$$\tau_t = \gamma_t, \ m_t = \nabla f_i(w_t), \ D_t = 0, \ B_t = I.$$

The explicit method equation 2 is the solution of the following implicit problem:

$$w_{t+1}, \alpha_{t+1} = \underset{w \in \mathbb{R}^d, \alpha \in \mathbb{R}}{\arg\min} \frac{1}{2}\|w - w_t\|_2^2 + \gamma_t \alpha,$$

$$\text{s.t.} \quad f_i(w_t) + \langle \nabla f_i(w_t), w - w_t \rangle \leq \alpha. \tag{7}$$

The proof is presented in Appendix B.1. Note, that normally $\alpha$ is an upper bound for $f_i^*$. Hence, if $f_i^*$ is known, we can fix $\alpha = f_i^*$. This leads us to the Stochastic Polyak step-size method.

**Stochastic Polyak step-size (SPS).** The update rule for Stochastic Gradient Descent with Polyak step-size can be derived as follows:

We set parameters[a] as follows:

$$\alpha = f_i^*, \; m_t = \nabla f_i(w_t), \; D_t = 0, \; B_t = I,$$

and solve the following problem:

$$w_{t+1} = \arg\min_{w \in \mathbb{R}^d} \frac{1}{2}\|w - w_t\|_2^2, \text{ s.t. } f_i(w_t) + \langle \nabla f_i(w_t), w - w_t \rangle \leq f_i^*. \tag{8}$$

---

[a]Note that in this formulation, we do not optimize over $\alpha$, and therefore, the value for $\tau$ is not required. In the subsequent text, we will omit specifying a value for this parameter wherever it is unnecessary.

We demonstrate in Appendix B.2 that equation 3 serves as an explicit formulation of equation 8. When $f_i^*$ is known (as in the case of interpolation under Assumption 1), the method becomes both adaptive and parameter-free. Otherwise, an estimate of $f_i^*$ must be tuned, analogous to tuning the step-size parameter $\gamma_t$ in SGD. Furthermore, we show that a similar transition can be applied to other methods.

**Preconditioned SGD.** Preconditioning is used to introduce curvature information into SGD equation 5. We precondition the stochastic gradient approximation, denoted as $m_t$, with a positive definite matrix $B_t \succ 0$. There are many methods that fit this description, ranging from the classical Damped Newton method and Quasi-Newton methods (like BFGS) to modern diagonal preconditioning techniques such as Adam, AdaGrad, and Hutchinson method. We can derive Preconditioned SGD from equation 6.

With $0 < \gamma_t \leq 1$ as a step-size, we choose the parameters as follows:

$$\tau_t = \gamma_t, \; m_t = g_t, \; D_t = 0, \; B_t = B_t,$$

and solve the following problem:

$$w_{t+1}, \alpha_{t+1} = \arg\min_{w \in \mathbb{R}^d, \alpha \in \mathbb{R}} \frac{1}{2}\|w - w_t\|_{B_t}^2 + \gamma_t \alpha, \quad \text{s.t. } f_i(w_t) + \langle g_t, w - w_t \rangle \leq \alpha. \tag{9}$$

We get the next explicit step: $w_{t+1} = w_t - \gamma_t B_t^{-1} g_t$. Note that $g_t$ can represent either $\nabla f_i(w_t)$ or an alternative approximation of the gradient. This notation will also be used in the subsequent text.

Next, we describe some preconditioning methods.

**AdaGrad** is an adaptive optimization method that approximates the Hessian of the objective function using the cumulative squared gradient information to scale the learning rates. Accumulation of all previous gradients in the preconditioner $B_t$ leads to decay in the learning rate $\gamma_t$ which increases performance for sparse settings (non-frequent features) at the cost of degrading in case of dense settings.

The AdaGrad preconditioning is derived by: $m_t = g_t = \nabla f_i(w_t)$, and $B_t = \text{diag}\left(\sqrt{\sum_{j=1}^t g_j^2}\right)$.

**Adam** is incorporating both adaptive learning rates and momentum. The update rule involves the computation of the moving average of both the first and second moments of the gradients. The first moment ($\beta_1$) is the mean of the gradients, and the second moment ($\beta_2$) is the uncentered variance of the gradients.

The Adam preconditioning is derived by:

$$m_t = \frac{(1 - \beta_1)\sum_{j=1}^t \beta_1^{t-i} g_j}{1 - \beta_1^t}, \quad B_t = \text{diag}(\sqrt{\frac{(1 - \beta_2)\sum_{j=1}^t \beta_2^{t-j} g_j^2}{1 - \beta_2^t}}),$$

where $0 < \beta_1, \beta_2 < 1$ are two hyperparameters referred to as first and second moment coefficients. The biased estimates are corrected by dividing them by the bias correction terms, which are powers of the decay rates $\beta_1$ and $\beta_2$, respectively.

**Hutchinson's method** is employed to estimate the diagonal of the Hessian matrix (Hutchinson, 1989). To achieve this, the method utilizes only a handful of Hessian-vector products, which can be efficiently computed

using backpropagation (Christianson, 1992). Specifically, the product of a Hessian matrix $\nabla^2 f(w)$ and a vector $h$ can be computed through a directional derivative of the gradient, given by $\frac{d}{dt}\nabla f(w + th)|_{t=0} = \nabla^2 f(w)h$. Hutchinson's method leverages Hessian-vector products to estimate the diagonal through $\mathrm{diag}(\nabla^2 f(w)) = \mathbb{E}[h \odot (\nabla^2 f(w)h)]$, where $h$ is a random vector with Rademacher distribution[2] or a normal distribution as discussed in (Bekas et al., 2007) and Lemma B.4 in Appendix. Utilizing this identity, we can estimate the Hessian diagonal by a weighted average of each iteration's result: $B_t = \beta B_{t-1} + (1 - \beta)\,\mathrm{diag}(h \odot \nabla^2 f_{i_t}(w_t)h)$, where $\beta \in (0, 1)$ is a momentum parameter, $i_t$ is a number of a random function on the step $t$, and $B_0 = \frac{1}{k}\sum_{j=1}^k \mathrm{diag}(h_j \odot \nabla^2 f_j(w_0)h_j)$, where $k$ is a number of functions for initialization of the approximation. To ensure $B_t$ remains positive definite, especially in the face of potential non-convexities in the loss functions, we apply truncation by positive number $\mu$ and retain only the absolute values of elements given by $(B_t)_{j,j} = \max\{\mu, |B_t|_{j,j}\}$. Some of the recent works utilizing this method are PSPS (Abdukhakimov et al., 2023), Sophia (Liu et al., 2024), OASIS (Jahani et al., 2022), and others (Sadiev et al., 2022; Pirau et al., 2023).

**Preconditioned SPS.** Similarly to SGD and SPS, Polyak step-size could be introduced for Preconditioned SGD methods. Preconditioned SPS (PSPS) was presented by Abdukhakimov et al. (2023). It can be also derived from SANIA for $B_t \succ 0$.

We choose the parameters as follows:

$$\alpha = f_i^*,\ m_t = g_t,\ D_t = 0,\ B_t = B_t,$$

and solve the following problem:

$$w_{t+1} = \arg\min_{w \in \mathbb{R}^d} \frac{1}{2}\|w - w_t\|_{B_t}^2,\ \text{s.t. } f_i(w_t) + \langle m_t, w - w_t \rangle \le f_i^*. \tag{10}$$

We get the next explicit step:

$$w_{t+1} = w_t - \frac{f_i(w_t) - f_i^*}{\|m_t\|_{B_t^{-1}}^2}B_t^{-1}m_t. \tag{11}$$

---

**Theorem 1**

Let $f_i(w)$ be a convex, $L_{\max}$-Lipschitz smooth function that satisfy the *Interpolation Condition* (Assumption 1) for all $i \in \{1, \ldots, n\}$. Assume $B_t \succ 0$ is a sequence of positive definite matrices for all $t \in \{0, \ldots, T\}$, with $m_t = \nabla f_i(w_t)$, and that $B_t$ satisfies the ordering $B_t \succeq B_{t+1} \succeq \nu$ for some $\nu > 0$. Then, for the sequence $w_t$ generated by equation 11, the average iterate $\hat{w}_T = \frac{1}{T}\sum_{t=0}^{T-1} w_t$ satisfies the following convergence guarantee:

$$\mathbb{E}[f(\hat{w}_T) - f^*] \le \frac{2L_{\max}\|w_0 - w^*\|_{B_0}^2}{\nu T}. \tag{12}$$

---

In PSPS, the norm in the projection is changed to a weighted norm based on the preconditioning matrix $B_t \succ 0$, it helps to improve the convergence rate in case of badly scaled/ill-conditioned datasets.

**Gradient regularized Newton method.** One of the main issues of Newton method is a lack of global convergence. To solve it with provably fast convergence, Cubic Regularized Newton method was proposed by Nesterov & Polyak (2006). Later, to simplify subproblem solution, the gradient regularization was proposed by Mishchenko (2023); Doikov & Nesterov (2023). Next, we present a formulation of a Stochastic Cubic Newton Method with gradient regularization from equation 6.

With $L_2$ as a Lipschitz-continuous constant for Hessian, we choose the parameters as follows:

$$\tau_t = \sqrt{\frac{3}{L_2\|g_t\|}},\ m_t = g_t = \nabla f_i(w_t),\ D_t = H_t = \nabla^2 f_i(w_t),\ B_t = I,$$

---

[2] $h_j \in \{-1, +1\}$ with equal probability.

and solve the following problem:

$$w_{t+1}, \alpha_{t+1} = \underset{w \in \mathbb{R}^d, \alpha \in \mathbb{R}}{\arg\min} \frac{1}{2}\|w - w_t\|_2^2 + \alpha\sqrt{\frac{3}{L_2\|g_t\|}} \tag{13}$$

$$\text{s.t. } f_i(w_t) + \langle g_t, w - w_t \rangle + \frac{1}{2}H_t[w - w_t]^2 \leq \alpha.$$

We get the next step:

$$w_{t+1} = w_t - (H_t + I\sqrt{\frac{L_2}{3}\|g_t\|})^{-1}g_t.$$

**SP2.** In (Li et al., 2023), the constraint of SPS equation 3 was extended for the second-order information, aimed at incorporating additional curvature information to accelerate the convergence rate. Next, we present the implicit formulation of SP2 under Assumption 1:

$$w_{t+1} = \arg\min_{w \in \mathbb{R}^d} \frac{1}{2}\|w - w_t\|^2, \quad \text{s.t.} \quad f_i(w_t) + \langle g_t, w - w_t \rangle + \frac{1}{2}H_t[w - w_t]^2 = 0. \tag{14}$$

The explicit formulation was presented only for generalized linear models.

In next sections, we will propose a variant of explicit solution for SP2 with connection to Cubic Newton.

### 2.3 Proposed methods

**Gradient regularized Newton method with Polyak step-size.** Similarly to SGD and SPS, we propose a new version of Cubic Newton method with Polyak step-size and its stochastic version. If $f_i^*$ is known for example in case of interpolation with Assumption 1, then the method is parameter-free. This result is new both in deterministic and stochastic cases. Similarly to SGD, we fix $\alpha = f_i^*$ in equation 13 and get the next method.

We choose the parameters as follows:

$$\alpha = f_i^*, \ \tau_t = \sqrt{\frac{3}{L_2\|g_t\|}}, \ m_t = g_t = \nabla f_i(w_t), \ D_t = H_t = \nabla^2 f_i(w_t), \ B_t = I,$$

and solve the following problem:

$$w_{t+1} = \arg\min_{w \in \mathbb{R}^d} \frac{1}{2}\|w - w_t\|_2^2, \text{ s.t. } f_i(w_t) + \langle g_t, w - w_t \rangle + \frac{1}{2}H_t[w - w_t]^2 \leq f_i^*. \tag{15}$$

The explicit step is formulated as follows:

$$w_{t+1} = w_t - (1 - \kappa_t)\left[\kappa_t I + (1 - \kappa_t)H_t\right]^{-1}g_t, \tag{16}$$

where $\kappa_t = 0$ if $f_i(w_t) - f_i^* > \frac{1}{2}\|g_t\|_{H_t^{-1}}^2$, otherwise $\kappa_t$ is computed by Cubic Newton-type line-search.

**SANIA Quasi-Newton for $B_t \succ 0$.** Similarly to PSPS equation 10, this approach covers AdaGrad, Adam, Hutchinson's method, Quasi-Newton methods with $B_t \succ 0$, and Newton method for convex functions with $H_t \succ 0$. The method is inspired by Affine-Invariant Cubic Newton from Hanzely et al. (2022). Note, the Hessian approximation $B_t$ is used both in the scaling of the objective norm and in the constraint model. We derive it from equation 6.

The parameters are chosen as follows:

$$\alpha = f_i^*, \ \tau_t = \gamma_t, \ m_t = g_t, \ D_t = B_t, \ B_t = B_t,$$

and solve the following problem:

$$w_{t+1} = \arg\min_{w \in \mathbb{R}^d} \frac{1}{2}\|w - w_t\|_{B_t}^2, \text{ s.t. } f_i(w_t) + \langle m_t, w - w_t \rangle + \frac{1}{2}B_t[w - w_t]^2 \leq f_i^*.$$

The explicit step is:

$$w_{t+1} = w_t - \lambda_t B_t^{-1} m_t, \tag{17}$$

where for $v_t = \frac{2(f_i(w_t) - f_i^*)}{\|m_t\|_{B_t^{-1}}^2}$, we define

$$\lambda_t = \begin{cases} 1 - \sqrt{1 - v_t}, & \text{if } v_t \leq 1, \\ 1, & \text{otherwise.} \end{cases} \tag{18}$$

Note, that for $v_t > 1$, there is no solution of equation 2.3 and we define $\lambda_t = 1$ as a minimum of the constraint. The main difference between PSPS equation 11 and SANIA-Quasi-Newton equation 17 is the parameter $\lambda_t$. For equation 17, step-size $\lambda_t \leq 1$ in equation 18, while in contrast for equation 11 $\lambda_t$ could be much bigger than 1. For Newton method, the step-size $\lambda_t$ is naturally bounded by 1, which makes SANIA-Quasi-Newton step-size safer than the step-size of PSPS. More details, comparisons, and theoretical results are presented in Appendix.

---

**Lemma 1**

Let $f_i(x)$ be a convex function for all $i \in [1, \dots, n]$ and have the same minimum $w^*$ (Assumption 1), $B_t \succ 0$ are positive definite matrices for $t \in [0, \dots, T]$, and $m_t = \nabla f_i(w_t)$. Then for equation 17 method with the step size $\lambda_t \in (0, v_t)$, we have $\|w_{t+1} - w^*\|_{B_t}^2 < \|w_t - w^*\|_{B_t}^2$. Additionally, for $\lambda_t = v_t/2$, we get $\|w_{t+1} - w^*\|_{B_t}^2 \leq \|w_t - w^*\|_{B_t}^2 - (f_i(w_t) - f_i^*)v_t/2$.

---

**SANIA AdaGrad-SQR.** We propose a new preconditioning method, called AdaGrad-SQR, by removing the square root from AdaGrad update. In Section 2.4, we will prove that the improved algorithm have "scale invariance" property. Figure 1 shows that the proposed algorithm behaves the same both on original and scaled versions of datasets.

We define $m_t, B_t, D_t$ for equation 17 as follows:

$$m_t = g_t, \quad B_t = D_t = \text{diag}\left(\sum_{j=1}^t g_j^2\right). \tag{19}$$

**SANIA Adam-SQR.** Along with SANIA AdaGrad-SQR, we propose another "scale-invariant" method. Following the same idea, it removes the square root from the preconditioning matrix of Adam.

We define $m_t, B_t, D_t$ for equation 17 as follows:

$$m_t = \frac{(1 - \beta_1)\sum_{j=1}^t \beta_1^{t-j}g_j}{1 - \beta_1^t}, \quad B_t = D_t = \text{diag}\left(\frac{(1 - \beta_2)\sum_{j=1}^t \beta_2^{t-j}g_j^2}{1 - \beta_2^t}\right). \tag{20}$$

**SANIA PCG for Newton method for non-convex functions.** In cases where the functions $f_i(w)$ are non-convex, the Hessian matrix $H_t$ may not be positive definite but invertible. This characteristic renders the approach not applicable, as $\|g_t\|_{H_t^{-1}}$ is no longer a norm. To address this issue, we propose a solution based on the rank-1 SR-1 approximation.

First, let us define $B_t$ and $D_t$ as follows:

$$B_t = D_t = \frac{yy^\top}{s^\top y}, \ m_t = g_t, \ \alpha = f_i^*, \ \tau_t = \gamma_t, \text{ where } s = H_t^{-1}g_t \text{ and } y = H_t s = g_t.$$

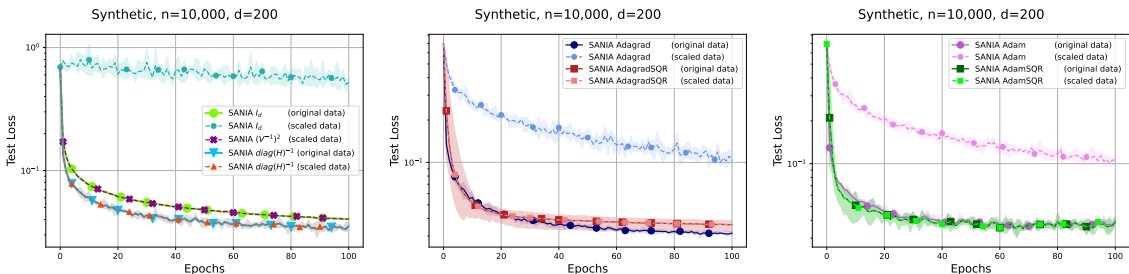

Figure 1: Observation of scale invariance of SANIA while minimizing *logistic regression* objective function on synthetic binary classification dataset with scaling factor $k = 4$.

Then, by solving the problem equation 2.3, we get an explicit method:

$$w_{t+1} = w_t - \lambda_t B_t^+ \nabla f_i(w_t),$$

where for $v_t = \frac{2(f_i(w_t) - f_i^*)}{\|g_t\|_{B_t^+}^2}$ we define $\lambda_t = \begin{cases} 1 - \sqrt{1 - v_t}, & \text{if} \quad v_t \leq 1, \\ 1, & \text{otherwise.} \end{cases}$

Note that $B_t$ is a rank-1 matrix, hence non-invertible, but it does have a pseudoinverse which is given by $B_t^+ = \frac{ss^\top}{s^\top y}$, hence, $B_t^+ g_t = H_t^{-1} g_t$.

We present more details in Appendix C.6. In practice, we solve $H_t^{-1} g_t$ by using Conjugate Gradient method, which allows to compute only Hessian-vector products without computing and storing the full Hessian $H_t$.

## 2.4 Affine and scale invariance

The family of Stochastic Gradient Methods with Polyak step-size offers an update rule that alleviates the need of fine-tuning the learning rate of an optimizer. However, existing first-order algorithms, whether stochastic or deterministic, perform poorly on ill-conditioned datasets. One possible reason for this is their strong dependence on the chosen basis. This is why, in machine learning, it is common practice to normalize data, as it makes the optimization space and basis more amenable. In the case of generalized linear models (GLM), the choice of basis is directly linked to the handling of ill-conditioned datasets. Changing the basis leads to improvement of conditioning.

**Affine invariance** is one of the key features of the Newton method, which makes it basis-independent (Nesterov & Nemirovskii, 1994; Nesterov, 2018). Let $A \in \mathbb{R}^{d \times d}$ be a non-degenerate matrix. We consider function $\phi(y) = f(Ay)$. By affine transformation, we denote $f(w) \to \phi(y) = f(Ay), w \to A^{-1}y$. Now, we discuss what is affine invariant friendly and what is not. First of all, the local Hessian norm $\|h\|_{\nabla^2 f(w)}$ is affine-invariant: $\|z\|_{\nabla^2 \phi(y)}^2 = \langle \nabla^2 \phi(y) z, z \rangle = \langle A^\top \nabla^2 f(Ay) A z, z \rangle = \langle \nabla^2 f(w) h, h \rangle = \|h\|_{\nabla^2 f(w)}^2$. However, the norm $\|z\|_I^2$ is not affine invariant. Second of all, Damped Newton method is affine invariant (Lemma 5.1.1 (Nesterov, 2018)). It means that for the function $f(w)$ Damped Newton method with affine invariant step-size $\gamma_t$ generates $w_{t+1} = w_t - \gamma_t [\nabla^2 f(w_t)]^{-1} \nabla f(w_t)$. For a function $\phi(y)$, Damped Newton method generates $y_{t+1} = y_t - \gamma_t [\nabla^2 \phi(y_t)]^{-1} \nabla \phi(y_k)$. If $y_0 = A^{-1} w_0$, then $\forall t : y_t = A^{-1} w_t$. Essentially, we get a bijection between $y_t$ and $w_t$. Also, the function values during the optimization are the same $\phi(y_t) = f(w_t)$. It means that for GLM, we will automatically get the best basis. Finally, we can show that SANIA Newton and SANIA CG are affine invariant, because the step-size $\lambda_t$ in equation 18 is affine-invariant friendly. All proofs are presented in Appendix D.2.

**Scale invariance** is a special case of affine invariance, where the matrix $A$ is a diagonal matrix. This implies the removal of rotations from the transformations, allowing only diagonal transformations. To distinguish scale invariance from affine invariance, we denote the transformation $V \in \mathbb{R}^{d \times d}$ as a non-degenerate diagonal matrix. It's evident that the diagonal preconditioning from AdaGrad, Adam, and Hutchinson is not affine invariant because it does not adapt to rotations. However, they could be scale invariant. It turns out that

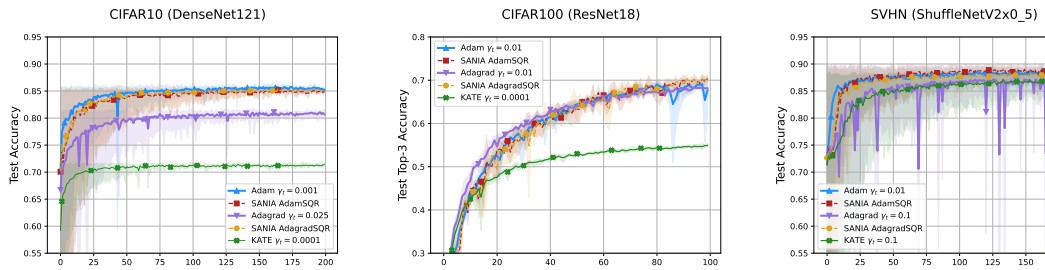

Figure 2: Performance of SANIA variants of Adam, Adagrad compared to standard Adam, Adagrad and KATE.

classical AdaGrad and Adam are not scale invariant, but if we remove the square root, they become scale invariant. We propose the new scale invariant SANIA AdaGrad-SQR in equation 19 and new scale invariant SANIA Adam-SQR in equation 20. All proofs are presented in Appendix D.2. Scale invariance property of SANIA Adam-SQR and SANIA AdaGrad-SQR is shown in Figure 1, where SANIA Adam-SQR and SANIA AdaGrad-SQR are converging identically for both original and badly scaled versions of the datasets, while using classical Adam and AdaGrad preconditioners result in different convergence steps. Recently, scale invariant version of AdaGrad, named KATE, was proposed by Choudhury et al. (2024).

Figure 1 illustrates that SANIA is able to become scale invariant with various preconditioners. Note that SANIA $B_t = I_d$, SANIA $B_t = \text{diag}((V^{-1})^2)$, and SANIA $B_t = \text{diag}(H^{-1})$ are preconditioned by Identity matrix (i.e. no preconditioning), squared inverse of the scaling vector used to obtain the scaled version of the dataset, and inverse of the Hessian diagonal of the objective function, respectively. One of the most noteworthy observations from this figure is that using the vector employed to transform the dataset for scaling, as a preconditioner, results in a scale invariant method. This essentially leads to convergence in a similar manner as non-preconditioned SANIA applied to the original dataset. In practice, obtaining such information is typically unattainable and often not even approximable. However, by utilizing the curvature of the objective function, we can achieve the same scale invariance property. This is also demonstrated in Figure 1 by comparing SANIA preconditioned with the diagonal of the Hessian (SANIA $\text{diag}(H_t^{-1})$) on both the original and scaled data. This method results in improved convergence while maintaining scale invariance, albeit with minor numerical instabilities. Nevertheless, SANIA $\text{diag}(H_t^{-1})$ is still impractical for large problems involving demanding calculations of Hessian. For reference, in the same figure we display performance of Adam with a constant step size, which deteriorates when scaled data is introduced.

## 3 Experiments

We test our methods on multiclass and binary classification problems with both linear models and neural networks. Considering practicality of the methods in experiments we only focus on SANIA Adam-SQR and SANIA Adagrad-SQR. For experiments with NNs we choose 5 architectures, namely **LeNet5** Lecun et al. (1998), **Simple Convolutional Neural Network** with 2 convolutional layers ($\sim 400K$ parameters), **DenseNet121** Huang et al. (2018), **ResNet18** He et al. (2015) and **ShuffleNetV2** with 0.5x output channels Ma et al. (2018a) trained on 5 datasets, **MNIST** LeCun et al. (2010), **Fashion-MNIST** Xiao et al. (2017), **CIFAR10** and **CIFAR100**, Krizhevsky et al. (2009) and **SVHN** Netzer et al. (2011) respectively. For evaluations with a linear model on binary classification problems we consider **logistic regression** that is defined as $f_{LogReg}(w) = \frac{1}{n} \sum_{i=1}^{n} \log(1 + \exp(-y_i x_i^T w))$, where $\{(x_i, y_i)\}_{i=1}^n$ is our dataset, $x_i \in \mathbb{R}^d$ and $y_i \in \{-1, +1\}$. We select small and large scale datasets from **LibSVM** data repository lib and conduct additional experiments to illustrate performance and scale invariance property of out methods. To simulate badly scaled data we introduce *scaled* version of each dataset where its feature columns are multiplied by a vector $e = \{\exp(a_i)\}_{i=1}^d$ where $a_i$ is generated from a uniform distribution on the interval $[-k, k]$.

All experiments are conducted with 5 initial seeds (0-4) and learning rates for Adam and Adagrad are chosen after multiple rounds of manual fine-tuning. Additional experiments, findings and other details (synthetic dataset generation, learning rates and etc.) can be found in Appendix E. The source code is available at `https://anonymous.4open.science/r/SANIA-A12E`.

In Figure 2 (see also Figures 3 and 6 in appendix) we can see that all presented variations of SANIA closely match or outperform other adaptive optimization methods across both under- and over-parametarized settings. Once again, note that while other methods require step-size fine-tuning and multiple runs of experiments, SANIA only needs one run for one set of configurations (i.e. scaling factor, batch-size, and etc.).

## 4    Conclusion

In this paper, we introduced a versatile and inclusive framework that not only encompasses classical optimization techniques but also sheds valuable light on Polyak step-size methods. Our research introduce the first Cubic Newton method with Polyak step-size which combines the efficiency of stochastic methods and the robustness of Newton methods. We have presented innovative variants of AdaGrad and Adam optimization algorithms that are scale invariant. Our proposed methods are affine or scale invariant, and this important development ensures the invariance of these methods to basis transformation, expanding their applicability and reliability in various scenarios. Our work is supported by comprehensive experiments including both convex and non-convex settings.

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

## A   RELATED WORK

Second-order methods have played a crucial role in contemporary optimization since their inception in classical works focused on root-finding algorithms by Newton (1687), Raphson (1697), Simpson (1740), and Bennett (1916). Subsequent significant advancements in the Newton method and its local quadratic convergence rates were made by Kantorovich (1948b;a; 1949; 1951b;a; 1956; 1957). These methods have been extensively researched, refined, and enhanced in various works, with notable contributions from Moré (1977),Griewank (1981), Nesterov & Polyak (2006). Today, they are widely employed in both industrial and scientific computing. For a comprehensive historical overview of the Newton method, Boris T. Polyak's paper Polyak (2007) provides more in-depth insights. Compared to first-order algorithms, second-order methods typically yield faster convergence. However, it's important to note that the per-iteration computational cost of second-order methods is considerably higher. An example of the classical Newton method can be expressed as follows:

$$x_{t+1} = x_t - \left[\nabla^2 f(x_t)\right]^{-1} \nabla f(x_t).$$

It exhibits quadratic local convergence, but it becomes impractical for large-scale optimization problems due to the necessity of computing the complete Hessian and matrix inversion at each iteration. It also lack of global convergence properties and could diverge if far from the solution.

The Cubic Regularized Newton method by Yurii Nesterov and Boris T. Polyak (Nesterov & Polyak, 2006) is one of the main approaches to globalize the Newton method. This algorithm achieves global convergence with the convergence rate $O(\varepsilon^{-1/2})$ for convex functions. Nonetheless, a notable limitation of the Cubic Regularized Newton method lies in the auxiliary problem, which typically requires running a separate optimization algorithm to solve it. Several research papers have proposed regularization techniques based on the gradient norm, aiming to derive an explicit regularized Newton step Polyak (2009; 2017). In Mishchenko (2023); Doikov & Nesterov (2023), the convergence rate was improved up to $O(\varepsilon^{-1/2})$ for convex functions, under higher assumptions on smoothness it accelerates up to $O(\varepsilon^{-1/3})$ Doikov et al. (2024). Affine-Invariant Cubic Regularized Newton method with local Hessian norms has the convergence rate $O(\varepsilon^{-1/2})$ and the same subproblem as a classical Newton step Hanzely et al. (2022).

## B   Proofs

### B.1   Stochastic Gradient Descent with SANIA

---

**Lemma 2**

The solution $\bar{w}$ of the next problem

$$\bar{w} = \underset{w \in \mathbb{R}^d, \alpha \in \mathbb{R}}{\arg\min} \ f(w) + \tau\alpha \quad s.t \ \ g(w) \leq \alpha \tag{21}$$

is the same as the solution $\hat{w}$ of

$$\hat{w} = \underset{w \in \mathbb{R}^d}{\arg\min} f(w) + \tau g(w), \tag{22}$$

where $\tau > 0$.

---

*Proof.* Denote the Lagrangian as $\mathcal{L}(w, \alpha, \lambda) = f(w) + \tau\alpha + \lambda(g(w) - \alpha)$, where $\lambda \geq 0$ is the Lagrange multiplier. We know that $\frac{\partial \mathcal{L}}{\partial \alpha} = \tau - \lambda$ should be 0, which means $\lambda = \tau > 0$. According to the complementary slackness, the condition $\lambda(g(w) - \alpha) = 0$ should hold. Thus, $\alpha = g(w)$, which means solving problem 21 is the same as solving problem 22. □

> **Lemma 3: Stochastic Gradient Descent**
>
> Let $\gamma_t > 0$, then the solution to
>
> $$w_{t+1}, \alpha_{t+1} = \underset{w \in \mathbb{R}^d, \alpha \in \mathbb{R}}{\arg\min} \frac{1}{2}\|w - w_t\|_2^2 + \gamma_t \alpha \quad \text{s.t. } f_i(w_t) + \langle \nabla f_i(w_t), w - w_t \rangle \le \alpha, \qquad (23)$$
>
> is given by
>
> $$w_{t+1} = w_t - \gamma_t \nabla f_i(w_t) \qquad (24)$$

*Proof.* From Lemma B.1, we know that solving problem 23 is the same as solving the following problem:

$$w_{t+1} = \underset{w \in \mathbb{R}^d}{\arg\min} \frac{1}{2}\|w - w_t\|_2^2 + \gamma_t(f_i(w_t) + \langle \nabla f_i(w_t), w - w_t \rangle). \qquad (25)$$

By taking the derivative of the objective function, we get the solution right away. $\qquad\square$

## B.2 Stochastic Polyak step-size with SANIA

> **Lemma 4: Stochastic Polyak step-size**
>
> $f_i^*$ is the minimal value of function $f_i(w_t)$. The solution to
>
> $$w_{t+1} = \underset{w \in \mathbb{R}^d}{\arg\min} \frac{1}{2}\|w - w_t\|_2^2 \quad \text{s.t. } f_i(w_t) + \langle \nabla f_i(w_t), w - w_t \rangle \le f_i^*, \qquad (26)$$
>
> is given by
>
> $$w_{t+1} = w_t - \frac{f_i(w_t) - f_i^*}{\|\nabla f_i(w_t)\|^2} \nabla f_i(w_t). \qquad (27)$$

*Proof.* Denote the Lagrangian as $\mathcal{L}(w, \lambda) = \frac{1}{2}\|w - w_t\|_2^2 + \lambda(f_i(w_t) + \langle \nabla f_i(w_t), w - w_t \rangle - f_i^*)$, and we can get Karush–Kuhn–Tucker(KKT) conditions as below:

$$\begin{cases} \frac{\partial \mathcal{L}}{\partial w} = w - w_t + \lambda \nabla f_i(w_t) = 0 \\ \lambda(f_i(w_t) + \langle \nabla f_i(w_t), w - w_t \rangle - f_i^*) = 0 \\ f_i(w_t) + \langle \nabla f_i(w_t), w - w_t \rangle - f_i^* \le 0 \\ \lambda \ge 0. \end{cases} \qquad (28)$$

$\lambda \in \mathbb{R}_+$ is called Lagrange multiplier, and if $\lambda = 0$, then the constrain is not active. We consider these two cases as following.

(i) $\lambda = 0$: $\begin{cases} w_{t+1} = w_t \\ f_i(w_t) - f_i^* \le 0, \textit{It's only true when they are equal.} \end{cases}$    (ii) $\lambda > 0$: $\begin{cases} w_{t+1} = w_t - \lambda \nabla f_i(w_t) \\ \lambda = \frac{f_i(w_t) - f_i^*}{\|\nabla f_i(w_t)\|^2}. \end{cases}$

$\qquad\square$

### B.3 Preconditioned SGD with SANIA

---

**Lemma 5: Preconditioned SGD**

Let $B_t \in \mathbb{R}^{d \times d}$ be a symmetric positive definite matrix. Let $\gamma_t > 0$, then the solution to

$$w_{t+1}, \alpha_{t+1} = \underset{w \in \mathbb{R}^d, \alpha \in \mathbb{R}}{\arg\min} \frac{1}{2}\|w - w_t\|_{B_t}^2 + \gamma_t \alpha \quad \text{s.t. } f_i(w_t) + \langle m_t, w - w_t \rangle \leq \alpha, \tag{29}$$

is given by:

$$w_{t+1} = w_t - \gamma_t B_t^{-1} m_t. \tag{30}$$

For **AdaGrad** setting, we let

$$m_t = \nabla f_i(w_t), \ B_t = \sqrt{\sum_{j=1}^t g_j \odot g_j};$$

for **Adam** setting,

$$m_t = \frac{(1 - \beta_1) \sum_{j=1}^t \beta_1^{t-j} g_j}{1 - \beta_1^t}, B_t = \sqrt{\frac{(1 - \beta_2) \sum_{j=1}^t \beta_2^{t-j} g_j \odot g_j}{1 - \beta_2^t}};$$

for **KATE** setting,

$$b_t = \sum_{j=1}^t g_j \odot g_j, \ m_t = \left( \sum_{j=1}^t \eta(g_j \odot g_j) + \frac{g_j \odot g_j}{b_j \odot b_j} \right) g_t, \ B_t = \mathrm{diag}(b_t);$$

and for **Sophia** setting,

$$m_t = \beta_1 m_{t-1} + (1 - \beta_1) g_t, \ \ B_t = \mathrm{Estimator}(w_t).$$

**Sophia** employs clipping, hence the update rule is slightly modified:

$$w_{t+1} = w_t - \gamma_t \cdot clip(B_t^{-1} m_t).$$

---

*Proof.* From Lemma B.1, we know problem 29 is equivalent to:

$$w_{t+1} = \underset{w \in \mathbb{R}^d}{\arg\min} \frac{1}{2}\|w - w_t\|_{B_t}^2 + \gamma_t (f_i(w_t) + \langle m_t, w - w_t \rangle). \tag{31}$$

Take derivative of $w$ and get solution:

$$w_{t+1} = w_t - \gamma_t B_t^{-1} m_t. \tag{32}$$

By plugging in $m_t$ and $B_t$, we get formulas for AdaGrad: $w_{t+1} = w_t - \gamma_t \dfrac{g_t}{\sqrt{\sum_{j=1}^t g_j \odot g_j}}$,

and for Adam: $w_{t+1} = w_t - \gamma_t \dfrac{\frac{(1-\beta_1) \sum_{j=1}^t \beta_1^{t-j} g_j}{1 - \beta_1^t}}{\sqrt{\frac{(1-\beta_2) \sum_{j=1}^t \beta_2^{t-j} g_j \odot g_j}{1 - \beta_2^t}}}$

$\square$

### B.4 Hutchinson's Lemma

---

**Lemma 6: Hutchinson**

Let $I \in \mathbb{R}^{d \times d}$ be the identity matrix. Let $H \in \mathbb{R}^{d \times d}$ and let $z \in \mathbb{R}^d$ be a random vector with a distribution such that

$$\mathbb{E}[zz^T] = I. \tag{33}$$

It follows that

$$\text{diagonal}(H) = \mathbb{E}[z \odot Hz]. \tag{34}$$

Furthermore if $z$ has Rademacher or a normal distribution, then 33 holds.

---

*Proof.* Taking expectation the Hadamard product we have that

$$\mathbb{E}[z \odot Hz] = \mathbb{E}[\sum_i z_i (\sum_j H_{ij} z_j) e_i] = \sum_i \sum_j H_{ij} \mathbb{E}[z_j z_i] e_i. \tag{35}$$

Since $\mathbb{E}[z_j z_i] = I$ we have that $\mathbb{E}[z_j z_i] = \delta_{ij} = \begin{cases} 1 & \text{if } i = j \\ 0 & \text{if } i \neq j. \end{cases}$

Using the above in 35 we have that

$$\mathbb{E}[z \odot Hz] = \sum_i H_{ii} e_i \tag{36}$$

which is the diagonal of the Hessian matrix.

Let $z$ be a Rademacher random varaible. That is $z_i = \begin{cases} 1 & \text{with probability } \frac{1}{2} \\ -1 & \text{with probability } \frac{1}{2}. \end{cases}$ Thus for $i, j \in 1, \ldots, d$ and $i \neq j$, we have that $\mathbb{E}[z_i] = 0$, $\mathbb{E}[z_i^2] = 1$ and $\mathbb{E}[z_i z_j] = 0$. The same results follow for $z \in \aleph(0, 1)$. $\qquad \square$

## C  Proposed methods

### C.1  Gradient regularized Newton method with Polyak step-size

---

**Lemma 7: Gradient regularized Newton method with Polyak step-size.**

$f_i^*$ is the minimal value of function $f_i(w_t)$. The solution to

$$w_{t+1} = \arg\min_{w \in \mathbb{R}^d} \frac{1}{2} \|w - w_t\|_2^2 \tag{37}$$

$$\text{s.t. } f_i(w_t) + \langle g_t, w - w_t \rangle + \frac{1}{2} H_t [w - w_t]^2 \leq f_i^*.$$

is given by

$$w_{t+1} = w_t - (1 - \kappa_t) \left[ (1 - \kappa_t) H_t + \kappa_t I \right]^{-1} g_t,$$

where $\kappa_t = 0$ if $f_i(w_t) - f_i^* > \frac{1}{2} \|g_t\|_{H_t^{-1}}^2$, otherwise $\kappa_t$ is a solution of the next equation

$$\mathcal{C}(\kappa) = f_i(w_t) - f_i^* - \frac{1 - \kappa}{2} g^\top \left[ (1 - \kappa) H_t + \kappa I \right]^{-1} g_t - \frac{\kappa(1 - \kappa)}{2} \left\| \left[ (1 - \kappa) H_t + \kappa I \right]^{-1} g_t \right\|_2^2 = 0,$$

which can be computationally solved by segment-search for $\kappa \in [0, 1]$. Note, that $\mathcal{C}(1) > 0$, and $\mathcal{C}(0) < 0$ if $f_i(w_t) - f_i^* \leq \frac{1}{2} \|g_t\|_{H_t^{-1}}^2$ hence the solution exists and could be found by bisection search.

---

*Proof.* For problem equation 37, the Lagrangian could be written as follows:

$$\mathcal{L}(w, \lambda) = \frac{1}{2}\|w - w_t\|_2^2 + \lambda \left( f_i(w_t) + \langle g_t, w - w_t \rangle + \frac{1}{2}H_t[w - w_t]^2 - f_i^* \right).$$

Then, we get the next KKT conditions:

$$\begin{cases} \frac{\partial \mathcal{L}}{\partial w} = I(w - w_t) + \lambda \left( g_t + H_t(w - w_t) \right) = 0 \\ \lambda(f_i(w_t) + \langle g_t, w - w_t \rangle + \frac{1}{2}H_t[w - w_t]^2 - f_i^*) = 0 \\ f_i(w_t) + \langle g_t, w - w_t \rangle + \frac{1}{2}H_t[w - w_t]^2 - f_i^* \le 0 \\ \lambda \ge 0. \end{cases}$$

Similarly, to previous proofs, the case of inactive constraint with $\lambda = 0$ us trivial and we focus on active constraint case.

$$\begin{cases} I(w - w_t) + \lambda \left( g_t + H_t(w - w_t) \right) = 0 \\ f_i(w_t) + \langle g_t, w - w_t \rangle + \frac{1}{2}H_t[w - w_t]^2 - f_i^* = 0, \\ \lambda > 0. \end{cases}$$

First, we find $w_{t+1}$ as

$$I(w - w_t) + \lambda \left( g_t + H_t[w - w_t] \right) = 0$$
$$w_{t+1} = w_t - \lambda \left[ \lambda H_t + I \right]^{-1} g_t.$$

Now, we substitute its new form in the active constraint and get

$$f_i(w_t) - f_i^* - \lambda g^\top \left[ \lambda H_t + I \right]^{-1} g_t + \frac{\lambda}{2} g^\top \left[ \lambda H_t + I \right]^{-1} \lambda H_t \left[ \lambda H_t + I \right]^{-1} g_t = 0$$

$$f_i(w_t) - f_i^* - \lambda g^\top \left[ \lambda H_t + I \right]^{-1} g_t + \frac{\lambda}{2} g^\top \left[ \lambda H_t + I \right]^{-1} (\lambda H_t + I) \left[ \lambda H_t + I \right]^{-1} g_t - \frac{\lambda}{2}\| \left[ \lambda H_t + I \right]^{-1} g_t \|_2^2 = 0$$

$$f_i(w_t) - f_i^* - \frac{\lambda}{2} g^\top \left[ \lambda H_t + I \right]^{-1} g_t - \frac{\lambda}{2}\| \left[ \lambda H_t + I \right]^{-1} g_t \|_2^2 = 0.$$

To simplify the line-search by $\lambda \in [0, +\infty]$, we transform it to $\kappa = \frac{1}{1+\lambda}$, which is now $\kappa \in [0, 1]$.

$$f_i(w_t) - f_i^* - \frac{1 - \kappa}{2} g^\top \left[ (1 - \kappa)H_t + \kappa I \right]^{-1} g_t - \frac{\kappa(1 - \kappa)}{2} \left\| \left[ (1 - \kappa)H_t + \kappa I \right]^{-1} g_t \right\|_2^2 = 0.$$

To simplify the multiple computations of the inverse matrix, one can apply SVD for $H_t$ and get the next simplified formulation:

$$H_t = U_t S_t U_t^\top$$
$$\left[ (1 - \kappa)H_t + \kappa I \right]^{-1} = \left[ (1 - \kappa)U_t S_t U_t^\top + \kappa U_t I U_t^\top \right]^{-1} = U_t \left[ (1 - \kappa)S_t + \kappa I \right]^{-1} U_t^\top$$
$$f_i(w_t) - f_i^* - \frac{1 - \kappa}{2} g^\top U_t \left[ (1 - \kappa)S_t + \kappa I \right]^{-1} U_t^\top g_t - \frac{\kappa(1 - \kappa)}{2} \left\| \left[ (1 - \kappa)S_t + \kappa I \right]^{-1} U_t^\top g_t \right\|_2^2 = 0$$
$$\tilde{g}_t = U_t^\top g_t$$
$$f_i(w_t) - f_i^* - \frac{1 - \kappa}{2} \tilde{g}_t^\top \left[ (1 - \kappa)S_t + \kappa I \right]^{-1} \tilde{g}_t - \frac{\kappa(1 - \kappa)}{2} \left\| \left[ (1 - \kappa)S_t + \kappa I \right]^{-1} \tilde{g}_t \right\|_2^2 = 0,$$

where $S_t$ is a diagonal matrix. Note, that this type of line-search is pretty common for Cubic Newton Methods. It adds only additional logarithmic inversions $O(\log \varepsilon^{-1})$ compared to classical Newton. $\qquad \square$

## C.2 SANIA Quasi-Newton

---

**Lemma 8: Projection Quadratic Inequality**

Let $B \in \mathbb{R}^{d \times d}$ be a symmetric positive definite matrix. Let $f_i(w_t) \geq 0$. The solution to

$$w_{t+1} = \arg\min_{w \in \mathbb{R}^d} \frac{1}{2}\|w - w_t\|_{B_t}^2 \tag{38}$$

$$\text{s.t. } f_i(w_t) + \langle m_t, w - w_t \rangle + \frac{1}{2}\|w - w_t\|_{B_t}^2 \leq 0. \tag{39}$$

is given by

$$w_{t+1} = w_t - \left(1 - \sqrt{1 - \frac{2(f_i(w_t) - f_i^*)}{\|m_t\|_{B_t^{-1}}^2}}\right) B_t^{-1} m_t, \tag{40}$$

if

$$\frac{2(f_i(w_t) - f_i^*)}{\|m_t\|_{B_t^{-1}}^2} \leq 1, \tag{41}$$

otherwise there is no feasible solution.

---

*Proof.* First we apply a change of coordinates and abbreviate. Let $x := B_t^{1/2}(w - w_t)$, $a := B_t^{-1/2}\nabla f_i(w_t)$ and $c := f_i(w_t)$. With this notation equation 38 is given by

$$\arg\arg\min_{x \in \mathbb{R}^d} \frac{1}{2}\|x\|^2 \text{ s.t. } \underbrace{c + \langle a, x \rangle + \frac{1}{2}\|x\|^2}_{=:q(x)} \leq 0. \tag{42}$$

The associated Lagrangian is given by

$$L(x, \mu) = \frac{1}{2}\|x\|^2 + \mu q(x),$$

where $\mu \geq 0$ is the Lagrange multiplier. Taking the derivative in $x$ and setting to zero gives

$$x = -\frac{\mu}{1 + \mu}a. \tag{43}$$

Consider the case that the constraint is not active, that is $\mu = 0$. Thus $x = 0$ and consequently $q(x) = c \geq 0$, which is only possible if the constraint is active thus a contradiction. Thus the constraint must be active and $\mu \neq 0$.

Let $\tau := \frac{\mu}{1+\mu}$. To determine $\tau$, and consequently $\mu$, we substituting back $x$ give in equation 43 into the constraint

$$q(x) = c - \tau\|a\|^2 + \frac{\tau^2}{2}\|a\|^2 = \left(1 - \sqrt{1 - \frac{2c}{\|a\|^2}} - \tau\right)\left(1 + \sqrt{1 - \frac{2c}{\|a\|^2}} - \tau\right)\frac{\|a\|^2}{2} = 0,$$

where we have factored $q(x)$ according to its roots in $\tau$. The above only has a solution if $1 - \frac{2c}{\|a\|^2} \geq 0 \Leftrightarrow \|a\|^2 \geq 2c$. In which case either root of $\tau$ is positive, but only the root $\tau = 1 - \sqrt{1 - \frac{2c}{\|a\|^2}}$ gives a positive $\mu$. Substituting this $\tau$ into equation 43 gives

$$x = -\left(1 - \sqrt{1 - \frac{2c}{\|a\|^2}}\right)a. \tag{44}$$

Substituting back $x := B_t^{1/2}(w - w_t)$, $a := B_t^{-1/2}\nabla f_i(w_t)$ and $c := f_i(w_t)$ gives

$$B_t^{1/2}(w_{t+1} - w_t) = -\left(1 - \sqrt{1 - \frac{2f_i(w_t)}{\|\nabla f_i(w_t)\|_{B_t^{-1}}^2}}\right) B_t^{-1/2}\nabla f_i(w_t). \tag{45}$$

Right multiplying by $B_t^{-1/2}$ and re-arranging gives the solution.

$\qquad\square$

### C.3 SANIA AdaGrad-SQR for Quasi-Newton.

The following is the explicit implementation of the Quasi-Newton algorithm when choosing AdaGrad-SQR as preconditioning matrix. We add some insurance $\epsilon$ to avoid numerical collapse.

---
**Algorithm 1** SANIA AdaGrad-SQR
---
Given batch size m, $\epsilon$, initial point $w \leftarrow 0$;
**for** $epoch = 0, 1, 2, \ldots$ **do**
    Set $G_0 = 0$
    **for** $t = 1, 2, \ldots$ **do**
        Compute gradient vector $g_t \leftarrow \frac{1}{m}\nabla_w \sum_{i=1}^m f_i(w)$     $f_i(w)$: stochastic objective function
        Accumulate $G_t \leftarrow G_{t-1} + g_t^2$
        $B_t = \text{diag}(G_t) + \epsilon$
        $\lambda_t \leftarrow$ step-size in equation 17
        $w \leftarrow w - \lambda_t B_t^{-1} g_t$
    **end**
**end**
---

### C.4 SANIA Adam-SQR for Quasi-Newton.

The following is the explicit implementation of the Quasi-Newton algorithm when choosing Adam-SQR as preconditioning matrix. We add some insurance $\epsilon$ to avoid numerical collapse.

---
**Algorithm 2** SANIA Adam-SQR
---
Given batch size m, $\epsilon, \beta_1, \beta_2$, initial point $w \leftarrow 0$;
**for** $epoch = 0, 1, 2, \ldots$ **do**
    Set $m_0 = 0, v_0 = 0$
    **for** $t = 1, 2, \ldots$ **do**
        $g_t \leftarrow \frac{1}{m}\nabla_w \sum_{i=1}^m f_i(w)$     Compute gradient vector
        $m_t \leftarrow \beta_1 m_{t-1} + (1 - \beta_1)g_t$   Accumulate $1^{st}$ momentum vector
        $v_t \leftarrow \beta_2 v_{t-1} + (1 - \beta_2)g_t^2$   Accumulate $2^{nd}$ momentum vector
        $\hat{m}_t \leftarrow m_t/(1 - \beta_1^t))$
        $\hat{v}_t \leftarrow v_t/(1 - \beta_2^t))$
        $B_t = \text{diag}(\hat{v}_t) + \epsilon$
        $\lambda_t \leftarrow$ step-size in equation 17
        $w \leftarrow w - \lambda_t B_t^{-1} \hat{m}_t$
    **end**
**end**
---

### C.5 SANIA PCG for Newton method on convex functions.

For convex setting where Hessian is positive definite, we can choose $B_t$ in equation 17 as Hessian or the approximation matrix of diagonal Hessian. We present detailed algorithm when $B_t = \nabla^2 f_i(w_t)$(we denote as $H_k$) as below.

---

**Algorithm 3** SANIA PCG for convex setting

---

Given $\epsilon, \gamma, \eta$, initial point $w \leftarrow 0$;
**for** *epoch* $= 0, 1, 2, \dots$ **do**

    **for** $k = 0, 1, 2, \dots$ **do**

        Set $s = 0, r_0 = \nabla f_k, z_0 = M_0^{-1} r_0, p_0 = z_0$    $\nabla f_k$ here is the stochastic mini-batch gradient as

        **for** $j = 0, 1, 2, \dots$ **do**

            $\alpha_j = \frac{r_j^T z_j}{p_j^T H_k p_j}$

            $s \leftarrow s + \alpha_j p_j$

            $r_{j+1} = r_j - \alpha_j H_k p_j$

            **if** $\|r_{j+1}\|_{M_k^{-1}} < \epsilon$ **then**

                **break**

            **end**

            $z_{j+1} = M_k^{-1} r_{j+1}$

            $\beta_j = \frac{r_{j+1}^T z_{j+1}}{r_j^T z_j}$

            $p_{j+1} = z_{j+1} + \beta_j p_j$

        **end**

        $\lambda_k \leftarrow$ step-size in equation 17

        $w \leftarrow w - \lambda_k s$

    **end**

**end**

---

In practice, we solve this matrix-vector product $(\nabla^2 f_i(w_t))^{-1} \nabla f_i(w_t)$ using Conjugate Gradient method. Furthermore, we can incorporate curvature information from Hessian approximation using Hutchinson's method, Adam or AdaGrad, which allows us to benefit from preconditioned system. In Conjugate Gradient method preconditioning is required to ensure faster convergence and the system can be preconditioned by a matrix $M^{-1}$ that is symmetric and positive-definite. Preconditioned Conjugate Gradient is equivalent to solving the following system:

$$E^{-1} \nabla^2 f_i(w_t)(E^{-1})^T E^T x = E^{-1} \nabla f_i(w_t),$$

where

$$EE^T = M.$$

If matrix $M_k = H_k$, then SANIA PCG is affine invariant; if $M_k = \text{diag}(H_k)$, then this method is scale invariant. In experiments you can choose $M_k$ as AdaGrad-SQR19 or Adam-SQR20.

### C.6 SANIA PCG for Newton method on non-convex functions.

For non-convex settings, we cannot use conjugate gradient method to solve this $Hx = g$ (Hessian is not positive definite) linear system of equations anymore. We try to combine Polyak step-size and line searxch Newton-CG method together to get good performance. The following is our specific implementation of the algorithm.

---

**Algorithm 4** SANIA PCG for Non-convex setting

---

Given $\epsilon, \gamma, \eta$, initial point $w \leftarrow 0$;

**for** $epoch = 0, 1, 2, \ldots$ **do**

    **for** $k = 0, 1, 2, \ldots$ **do**

        Set $s_0 = 0, x_0, r_0 = \nabla f_k, z_0 = M_0^{-1} r_0, p_0 = z_0$

        **for** $j = 0, 1, 2, \ldots$ **do**

            **if** $p_j^T H_k p_j \leq 0$ **then**

                $s_k = \gamma x_j + (1 - \gamma)\text{sign}(\nabla f_k^T p_j) p_j$

                $\lambda_k = \min(\frac{f_k}{\nabla f_k^T s_k}, \eta)$

                **break**

            **end**

            $\alpha_j = \frac{r_j^T z_j}{p_j^T H_k p_j}$

            $x_{j+1} = x_j + \alpha_j p_j$

            $r_{j+1} = r_j - \alpha_j H_k p_j$

            $z_{j+1} = M_k^{-1} r_{j+1}$

            **if** $r_{j+1}^T z_{j+1} < \epsilon$ **then**

                $s_k = x_{j+1}$

                $\lambda_k \leftarrow$ step-size in equation 17

                **break**

            **end**

            $\beta_j = \frac{r_{j+1}^T z_{j+1}}{r_j^T z_j}$

            $p_{j+1} = z_{j+1} + \beta_j p_j$

        **end**

        $w \leftarrow w - \lambda_k s_k$

    **end**

**end**

---

Since product $B_t^+ \nabla f_i(w_t)$ results in the same direction as $(\nabla^2 f_i(w_t))^{-1} \nabla f_i(w_t)$ , and now the algorithm stops once it detects negative curvature, otherwise it still takes CG steps until it hits stopping criteria. You can choose matrix $M_k$ to be AdaGrad-SQR19 or Adam-SQR20 to attain the scale-invariance property and we name them as SANIA PCG AdaGrad-SQR and SANIA PCG Adam-SQR. Notice that the names for the convex and non-convex setting are the same, but the implementation of these methods are slightly different due to the effectiveness of conjugate gradient methods.

## D   Affine and Scale Invariance

### D.1   Affine Invariance

---

**Lemma 9: Affine Invariance (Lemma 5.1.1 from (Nesterov, 2018))**

Let the sequence $\{x_k\}$ be generated by the Newton's method as applied to the function f:

$$x_{k+1} = x_k - [\nabla^2 f(x_k)]^{-1} \nabla f(x_k), \quad k \geq 0. \tag{46}$$

Consider the sequence $\{y_k\}$, generated by the Newton's method for the function $\phi$:

$$y_{k+1} = y_k - [\nabla^2 \phi(y_k)]^{-1} \nabla \phi(y_k), \quad k \geq 0, \tag{47}$$

with $y_0 = B^{-1} x_0$. Then $y_k = B^{-1} x_k$ for all $k \geq 0$.

---

*Proof.* Let $y_k = B^{-1}x_k$ for some $k \geq 0$. Then

$$
\begin{aligned}
y_{k+1} =& y_k - [\nabla^2 \phi(y_k)]^{-1}\nabla\phi(y_k) = y_k - [B^T\nabla^2 f(By_k)B]^{-1}B^T\nabla f(By_k) \\
=& B^{-1}x_k - B^{-1}[\nabla^2 f(x_k)]^{-1}\nabla f(x_k) = B^{-1}x_{k+1}.
\end{aligned}
$$

Thus, the Newton's method is affine invariant with respect to affine transformations of variables. $\qquad\square$

---

**Lemma 10: Affine Invariance for SANIA Newton**

Let the sequence $\{x_k\}$ be generated by the SANIA Newton method as applied to the function f:

$$
x_{k+1} = x_k - \lambda_k[\nabla^2 f(x_k)]^{-1}\nabla f(x_k), \quad k \geq 0. \tag{48}
$$

Consider the sequence $\{y_k\}$, generated by the SANIA Newton method for the function $\phi$:

$$
y_{k+1} = y_k - \hat{\lambda}_k[\nabla^2 \phi(y_k)]^{-1}\nabla\phi(y_k), \quad k \geq 0, \tag{49}
$$

with $y_0 = B^{-1}x_0$. Then $y_k = B^{-1}x_k$ for all $k \geq 0$.

---

*Proof.* We define

$$
\lambda_k = \begin{cases} 1 - \sqrt{1 - \upsilon_k}, & \text{if } \upsilon_k \leq 1, \\ 1, & \text{otherwise,} \end{cases} \tag{50}
$$

where

$$
\upsilon_k = \frac{2(f_i(x_k) - f_i^*)}{\|\nabla f_i(x_k)\|^2_{\nabla^2 f(x_k)^{-1}}} \tag{51}
$$

and

$$
\hat{\lambda}_k = \begin{cases} 1 - \sqrt{1 - \hat{\upsilon}_k}, & \text{if } \hat{\upsilon}_k \leq 1, \\ 1, & \text{otherwise,} \end{cases} \tag{52}
$$

where

$$
\hat{\upsilon}_k = \frac{2(\phi_i(y_k) - \phi_i^*)}{\|\nabla\phi_i(y_k)\|^2_{\nabla^2\phi(y_k)^{-1}}}. \tag{53}
$$

Let $y_k = B^{-1}x_k$ for some $k \geq 0$. We have this condition $\hat{\upsilon}_k = \frac{2(\phi_i(y_k) - \phi_i^*)}{\|\nabla\phi_i(y_k)\|^2_{\nabla^2\phi(y_k)^{-1}}} = \frac{2(f_i(By_k) - f_i^*)}{\|B^T\nabla f_i(By_k)\|^2_{[B^T\nabla^2 f(By_k)B]^{-1}}} = \frac{2(f_i(x_k) - f_i^*)}{\|\nabla f_i(x_k)\|^2_{\nabla^2 f(x_k)^{-1}}} = \upsilon_k$ holds, which means $\hat{\lambda}_k = \lambda_k$. Then

$$
\begin{aligned}
y_{k+1} =& y_k - \lambda_k[\nabla^2 \phi(y_k)]^{-1}\nabla\phi(y_k) = y_k - \lambda_k[B^T\nabla^2 f(By_k)B]^{-1}B^T\nabla f(By_k) \\
=& B^{-1}x_k - \lambda_k B^{-1}[\nabla^2 f(x_k)]^{-1}\nabla f(x_k) = B^{-1}x_{k+1}.
\end{aligned}
$$

Thus, the SANIA Newton method is affine invariant with respect to affine transformations of variables. $\quad\square$

### D.2  Scale Invariance

Kempka et al. (2019); Zhuang et al. (2022) illustrate this important but overlooked property of an optimization algorithm. It is widely recognized that the convergence rate of minimizing a twice continuously differentiable function $f$ through a first-order optimization algorithm depends heavily on the condition number. To mitigate the impact of the condition number, one effective approach is the use of preconditioners relying on Hessian of the function which yields affine invaraince. Consider the Hessian cannot be easily estimated, Zhuang et al. (2022) shows that scale invariance gives similar advantages to the use of an optimal diagonal preconditioner.

They also showed why algorithms like SGD and Adam have such excellent performances in DNNs even though they are not scale invariant. Because they are intensively linked to the use of batch normalization

which normalizes the gradients. Without BN, using SGD with momentum and weight decay, even a tiny learning rate will lead to divergence while training a deep neural network. But for the upgraded version of Adam–AdamW which enjoys scale invariance outperforms Adam when both are finely tuned.

Now, we will show the classical AdaGrad and Adam are not scale invariant but AdaGrad-SQR and Adam-SQR enjoy this property.

---

**Lemma 11: Scale Invariance of AdaGrad-SQR**

Let the sequence $\{x_k\}$ be generated by the AdaGrad-SQR as applied to the function f:

$$x_{k+1} = x_k - \lambda_k B_k^{-1} m_k, \quad k \geq 0, \quad \text{where } m_k = \nabla f_{i_k}(x_k), \ B_k = \sum_{j=1}^{k} \nabla f_{i_j}(x_j)^2. \tag{54}$$

Consider the sequence $\{y_k\}$, generated by the AdaGrad-SQR for the function $\phi$:

$$y_{k+1} = y_k - \hat{\lambda}_k \hat{B}_k^{-1} \hat{m}_k, \quad k \geq 0, \quad \text{where } \hat{m}_k = \nabla \phi_{i_k}(y_k), \ \hat{B}_k = \sum_{j=1}^{k} \nabla \phi_{i_j}(y_j)^2, \tag{55}$$

with $y_0 = V^{-1} x_0$. Then $y_k = V^{-1} x_k$ for all $k \geq 0$. $V$ is a diagonal matrix.

---

*Proof.* We define $\lambda_k = \begin{cases} 1 - \sqrt{1 - v_k}, & \text{if} \quad v_k \leq 1, \\ 1, & \text{otherwise}, \end{cases}$ where $v_k = \frac{2(f_i(x_k) - f_i^*)}{\|m_k\|_{B_k^{-1}}^2}$, and for $\hat{\lambda}_k$, $\hat{v}_k = \frac{2(\phi_i(y_k) - \phi_i^*)}{\|\hat{m}_k\|_{\hat{B}_k^{-1}}^2}$.

Let $y_k = V^{-1} x_k$ for some $k \geq 0$. We have

$$\begin{cases} \hat{B}_k = \sum_{j=1}^{k} \nabla \phi_{i_j}(y_k)^2 = \sum_{j=1}^{k} [V^T \nabla f_{i_j}(V y_k)]^2 = V^T [\sum_{j=1}^{k} \nabla f_{i_j}(x_k)^2] V = V^T B_k V, \\ \hat{m}_k = \nabla \phi_{i_j}(y_k) = V^T \nabla f_{i_j}(V y_k) = V^T \nabla f_{i_j}(x_k) = V^T m_k, \\ \hat{v}_k = \frac{2(\phi_i(y_k) - \phi_i^*)}{\|\hat{m}_k\|_{\hat{B}_k^{-1}}^2} = \frac{2(f_i(V y_k) - f_i^*)}{\|V^T m_k\|_{(V^T B_k V)^{-1}}^2} = \frac{2(f_i(x_k) - f_i^*)}{\|m_k\|_{B_k^{-1}}^2} = v_k. \end{cases}$$

Then

$$\begin{aligned} y_{k+1} &= y_k - \hat{\lambda}_k \hat{B}_k^{-1} \hat{m}_k = y_k - \lambda_k [V^T B_k V]^{-1} V^T m_k \\ &= V^{-1} x_k - \lambda_k V^{-1} B_k^{-1} m_k = V^{-1} x_{k+1}. \end{aligned}$$

Thus, the AdaGrad-SQR method is scale invariant.

And for Adam-SQR setting where $m_k = \frac{(1-\beta_1) \sum_{j=1}^{k} \beta_1^{k-j} \nabla f_{i_j}(x_k)}{1 - \beta_1^k}$, $B_k = \frac{(1-\beta_2) \sum_{j=1}^{k} \beta_2^{k-j} \nabla f_{i_j}(x_k)^2}{1 - \beta_2^k}$, and $\hat{m}_k = \frac{(1-\beta_1) \sum_{j=1}^{k} \beta_1^{k-j} \nabla \phi_{i_j}(y_k)}{1 - \beta_1^k}$, $\hat{B}_k = \frac{(1-\beta_2) \sum_{j=1}^{k} \beta_2^{k-j} \nabla \phi_{i_j}(y_k)^2}{1 - \beta_2^k}$. Similarly, we can get $\hat{B}_k = V^T B_k V$, $\hat{m}_k = V^T m_k$. Rest proofs are the same. From proofs above we can know for simple AdaGrad and Adam they are not scale invariant, because $\hat{B}_k = V^T B_k \neq V^T B_k V$. $\qquad \square$

### D.3 GLM

Suppose $f_i$ is the loss over a linear model with

$$f_i(w) = \psi_i(x_i^T w - y_i), \tag{56}$$

where $\psi_i : \mathbb{R} \to \mathbb{R}$ is the loss function, and $x_i$ is the $i^{th}$ data and $y_i$ is the corresponding label. Let the sequence $\{w_k\}$ be generated by method as applied to the function $f$. Consider the sequence $\{\hat{w}_k\}$, generated by the same method but for function $\phi$ where $\phi(\hat{w}_k) = f(B\hat{w}_k) = \psi_i(x_i^T B \hat{w}_k - y_i)$.

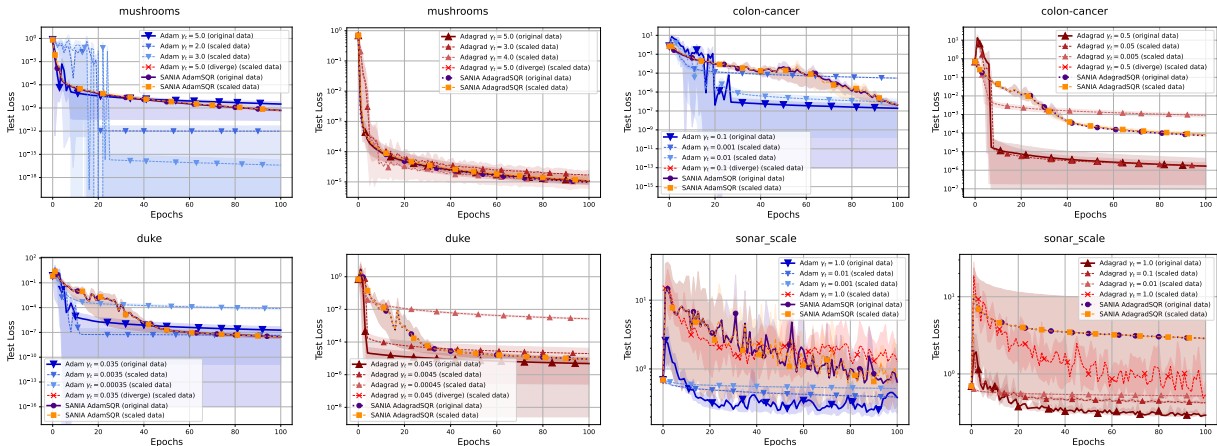

Figure 3: Observation of scale invariance of SANIA while minimizing *logistic regression* objective function on binary classification datasets from LibSVM with scaling factor $k = 4$.

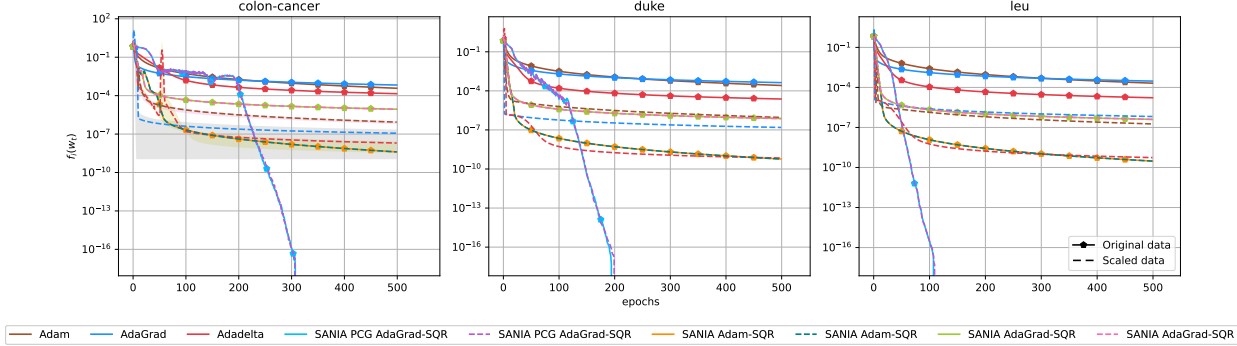

Figure 4: Performance of SANIA and other adaptive methods on 3 datasets (original and badly scaled with scaling factor $k = 6$) with *logistic regression* loss.

Take $x_i^T B$ as a whole, it can be seen as we are doing linear transformation to the data. When matrix $B$ is badly scaled, it will lead to a ill-conditioning dataset. And it inhibits the performance of the general algorithms, which is specifically reflected in the need for more iterations to converge, or even diverge on the worst case. But if the algorithm enjoys affine invariant property, that is, $\hat{w}_k = B^{-1}x_k$. Then we have $\psi_i(x_i^T B \hat{w}_k - y_i) = \psi_i(x_i^T B B^{-1} x_k - y_i) = f_i(w)$, which means we automatically have the same function value as the original one as every iteration goes.

# E   Additional Experiments and Details

All experiments were run with 5 different seeds $(0, 1, 2, 3, 4)$ using *PyTorch 2.0.1+cu118* on a computing machine with AMD EPYC 7402 24-Core Processor with 2.8GHz of base clock and 1 x NVIDIA RTX A6000 GPU unit. Default datatype in PyTorch is set to $\boxed{\text{torch.float64}}$. LibSVM[3] datasets and source code of optimizers used for the experiments are publicly available [4].

---

[3] https://www.csie.ntu.edu.tw/~cjlin/libsvmtools/datasets/
[4] https://anonymous.4open.science/r/SANIA-A12E

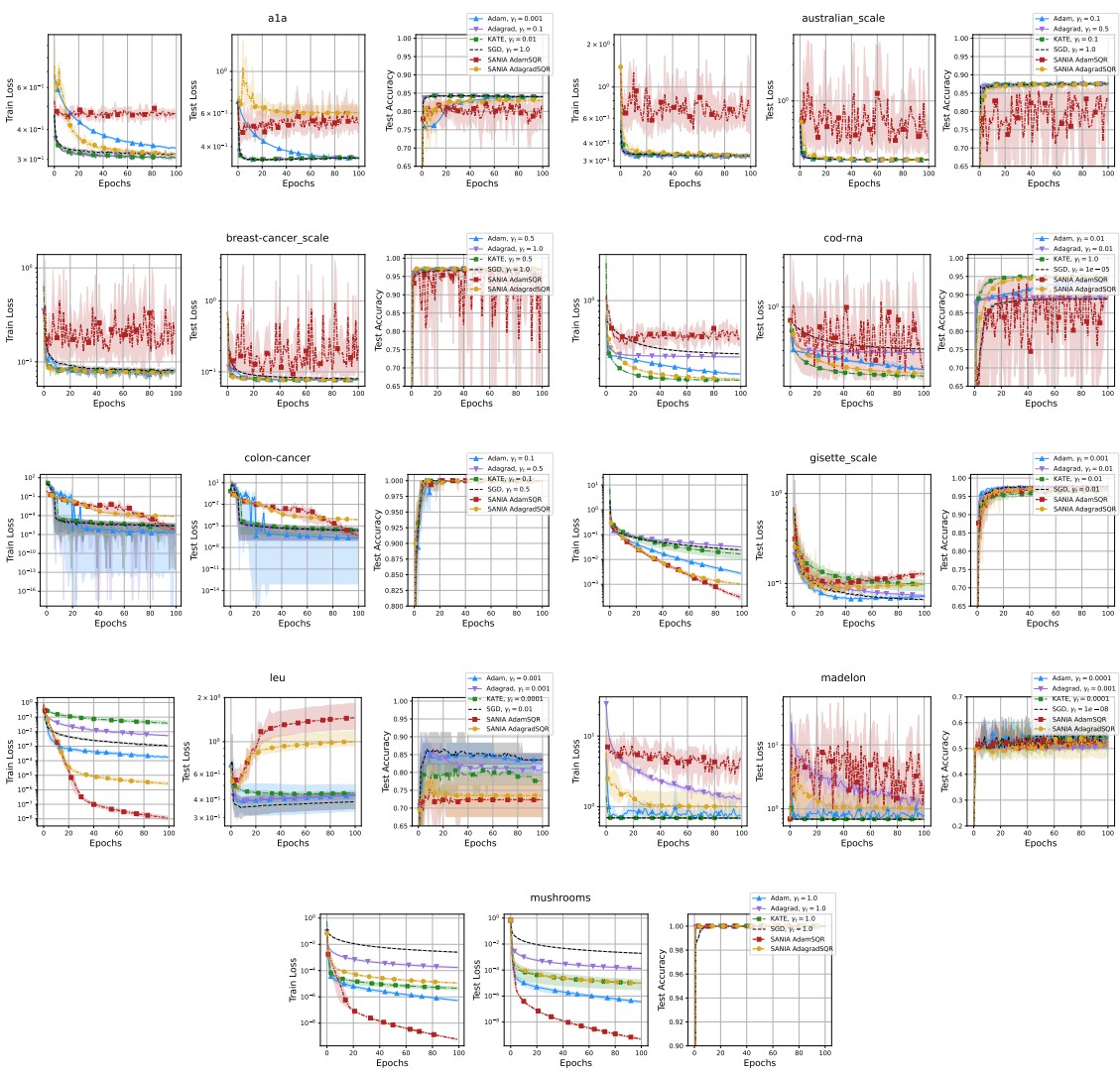

Figure 5: Performance of SANIA and other first-order optimization methods on binary classification tasks from LibSVM with *logistic regression* loss.

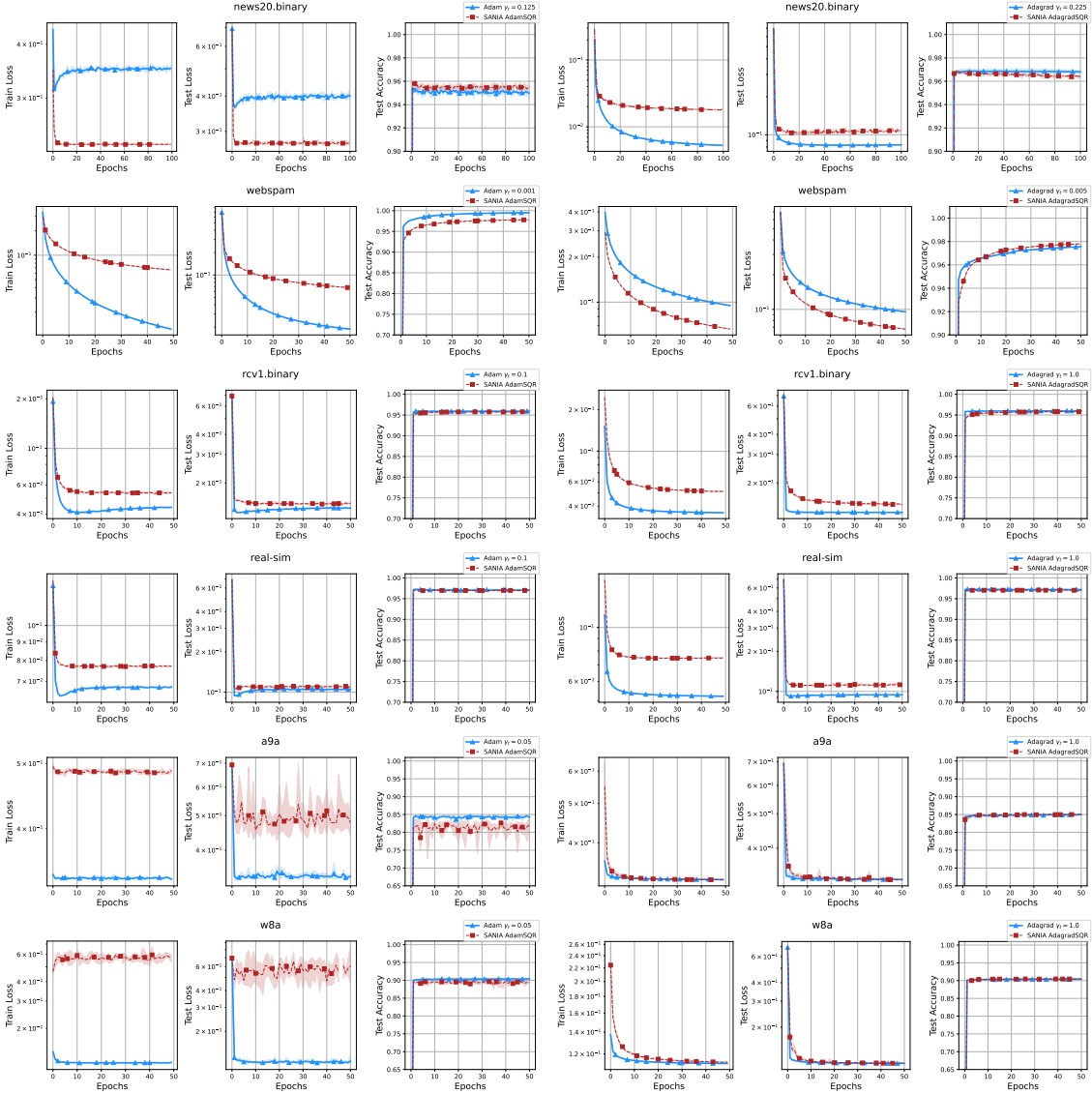

Figure 6: Large-scale binary classification experiments on datasets from LibSVM.

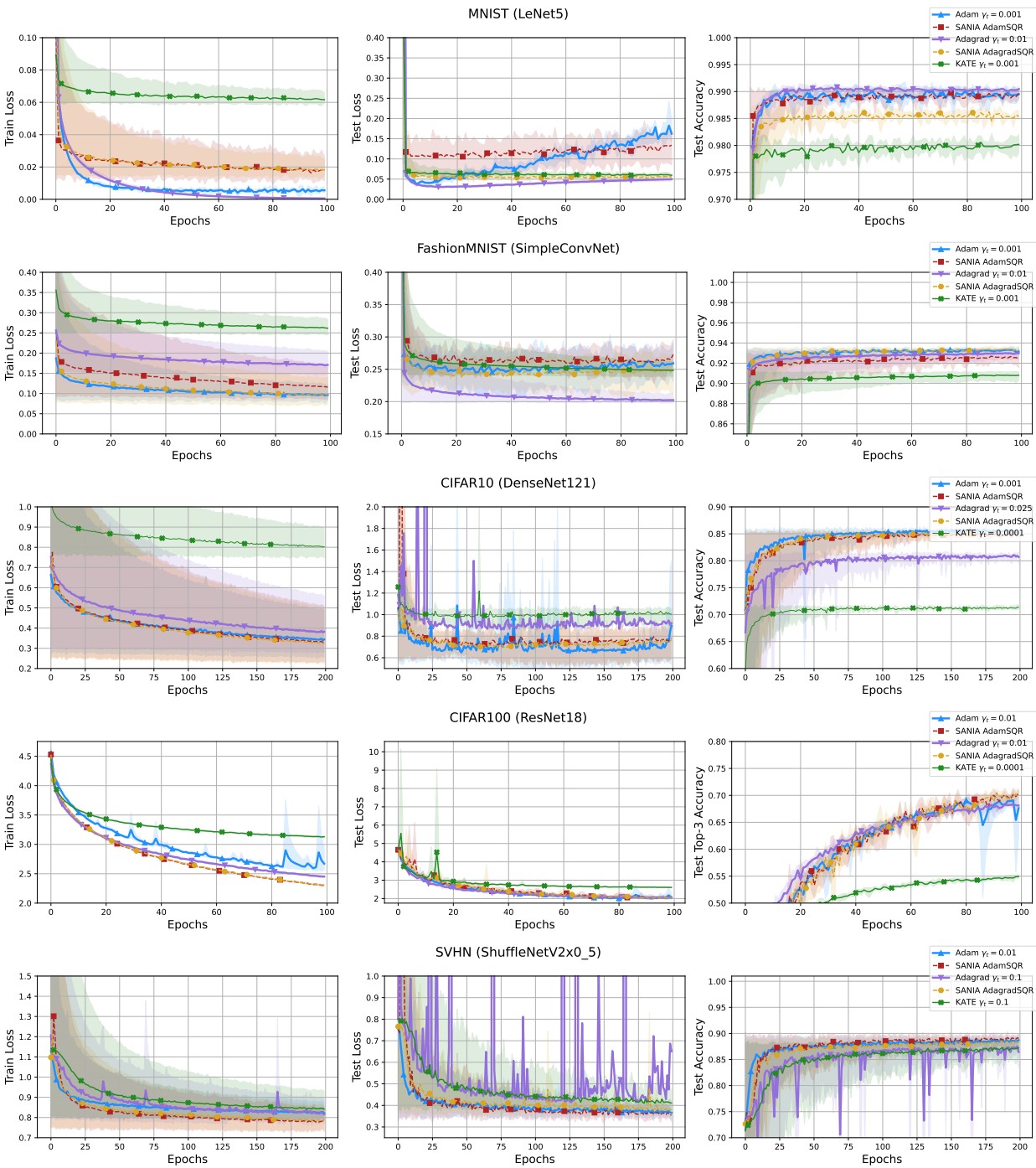

Figure 7: Performance of SANIA and other methods on multiple classification problems and neural networks.

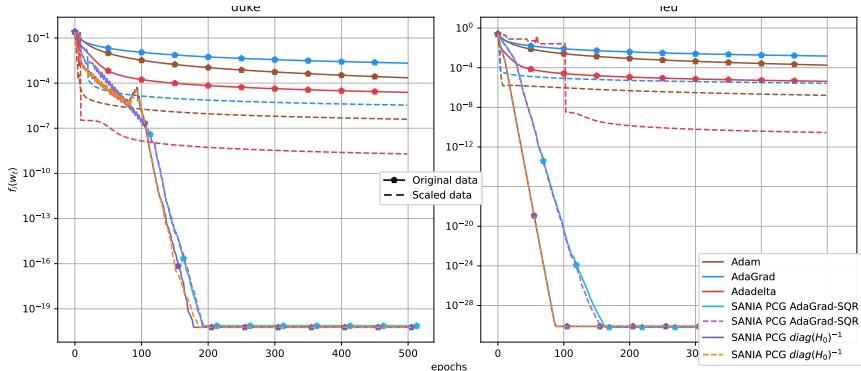

Figure 8: Performance of SANIA and other adaptive methods on 2 LibSVM datasets (original and badly scaled with scaling factor $k = 6$) with *non-linear least squares* loss.

### E.1 Non-linear least squares

To show experiments for non-convex problems, we use non-linear least squares in Figure 8. Let $\{(x_i, y_i)\}_{i=1}^n$ be our dataset, where $x_i \in \mathbb{R}^d$ and $y_i \in \{0, 1\}$, then *Non-linear least squares* problem is given by $f_{NLLSQ}(w) = \frac{1}{n} \sum_{i=1}^n (y_i - \frac{1}{(1+\exp(-x_i^T w))})^2$.

### E.2 Badly scaled dataset

In order to simulate badly scaled datasets we use scaling procedure shown in equation 57.

$$
A^{n \times d} = \begin{pmatrix} a_{1,1} & a_{1,2} & \dots & a_{1,d} \\ a_{2,1} & a_{2,2} & \dots & a_{2,d} \\ \vdots & \ddots & \ddots & \vdots \\ a_{n,1} & a_{n,2} & \dots & a_{n,d} \end{pmatrix} \xrightarrow{scale} \hat{A}^{n \times d} = \begin{pmatrix} a_{1,1} \times v_1 & a_{1,2} \times v_2 & \dots & a_{1,d} \times v_d \\ a_{2,1} \times v_1 & a_{2,2} \times v_2 & \dots & a_{2,d} \times v_d \\ \vdots & \ddots & \ddots & \vdots \\ a_{n,1} \times v_1 & a_{n,2} \times v_2 & \dots & a_{n,d} \times v_d \end{pmatrix},
\tag{57}
$$

where $v_i = e^{b_j}, \quad b_j \in \text{Uniform}[-k, k]$.

### E.3 Learning rates

Learning rates of algorithms used for experiments are **not** chosen randomly. To avoid overoptimized learning rates obtained using special algorithms and at the same time to adhere to some fairness of the results we conducted experiments with a series of learning rates $\gamma = 2^n$ where $n \in range(-2, -16, 2)$. Next, we used the best performing step size as the main result for certain optimizer.

### E.4 More findings

In Figure 9 we can see that proposed SANIA CG and SP2 for Generalized Linear Models presented in Li et al. (2023) generate identical steps towards the minimum given the exact same set of observations $x_i$. However, disadvantage of SP2 in this case is that it has a closed form solution only for GLMs.

Figure 11 shows that unlike other classical adaptive methods, SANIA with Newton step is scaling invariant. The same behaviour can be observed in Figure 12 where SANIA AdaGrad-SQR is not only scaling invariant but also displays significantly better performance compared to Adam, AdaGrad and Adadelta with a constant learning rate.

In Figure 10 we show how step-sizes of SANIA AdamSQR and SANIA AdagradSQR change during training on synthetic binary classification problem over 5 runs. Interestingly, evolution of step-sizes of SANIA AdamSQR closely resemble "warm-up" technique often used in practice, that is known to prevent instability in the beginning of training.

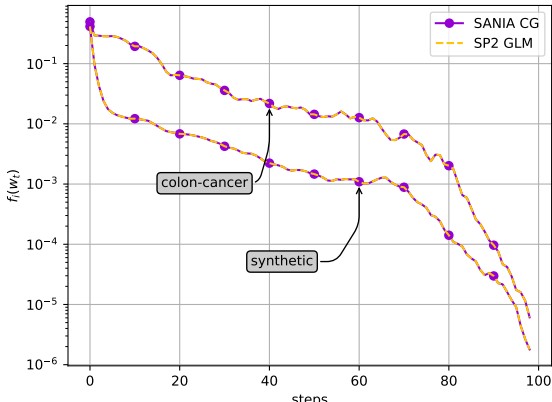

Figure 9: SANIA CG and SP2 GLM generate identical steps on *logistic regression* problem with batch size = 1.

### E.5 Experiments with Cubic Newton with Polyak step-size

In this subsection, we present results for Cubic Newton with Polyak step-size from equation 16. In Figure 13, we compare classical Cubic Newton from (Nesterov & Polyak, 2006), Gradient Regularized Newton from (Mishchenko, 2023; Doikov & Nesterov, 2023) and our Cubic Newton with Polyak step-size on full-batch logistic regression with $\frac{\mu}{2}\|w\|_2^2$-regularization, where $\mu = 1e - 4$. To show globalization properties, we choose the starting point far from the solution $x_0 = 3e$, where $e$ is a vector of all ones. We present Cubic Newton with theoretical parameter $L_2 = 0.1$, with fine-tuned parameter $L_2 = 0.0004$; Gradient Regularized Newton with fine-tuned parameter $L_2 = 0.0004$. There is a huge difference between fine-tuned and theoretical choice. It means that the method is pretty sensitive to the choice of the parameter $L_2$. For Cubic Newton with Polyak step-size, we denote approximate $f^*$ as $\hat{f}$. Then, we present the precise approximation $\hat{f} = f^* = 0.3361$, close lower approximation $\hat{f} = 0.3$, and the very simple and naive lower bound $\hat{f} = 0$. For all three cases, the convergence is almost the same. It also shows that Cubic Newton with Polyak step-size is very robust to the parameter $\hat{f}$, where even the most naive choice works perfectly fine. Finally, we highlight that Cubic Newton with Polyak step-size significantly overperform other Cubic methods even with fine-tuned parameters.

## F  Convergence Analysis

In this section, we prove the theoretical convergence results for SANIA Quasi-Newton and Preconditioned SPS (PSPS). These two methods are very close. Both of these methods have the next explicit form:

$$w_{t+1} = w_t - \lambda_t B_t^{-1} m_t \tag{58}$$

The difference is the step size. We introduce an additional parameter

$$v_t = \frac{2(f_i(w_t) - f_i^*)}{\|m_t\|_{B_t^{-1}}^2}. \tag{59}$$

For PSPS, the step size is

$$\lambda_t^{PSPS} = \frac{f_i(w_t) - f_i^*}{\|m_t\|_{B_t^{-1}}^2} = \frac{v_t}{2}. \tag{60}$$

For SANIA Quasi-Newton, the step size is

$$\lambda_t^{SANIA} = \begin{cases} 1 - \sqrt{1 - v_t}, & \text{if } v_t \leq 1, \\ 1, & \text{otherwise.} \end{cases} \tag{61}$$

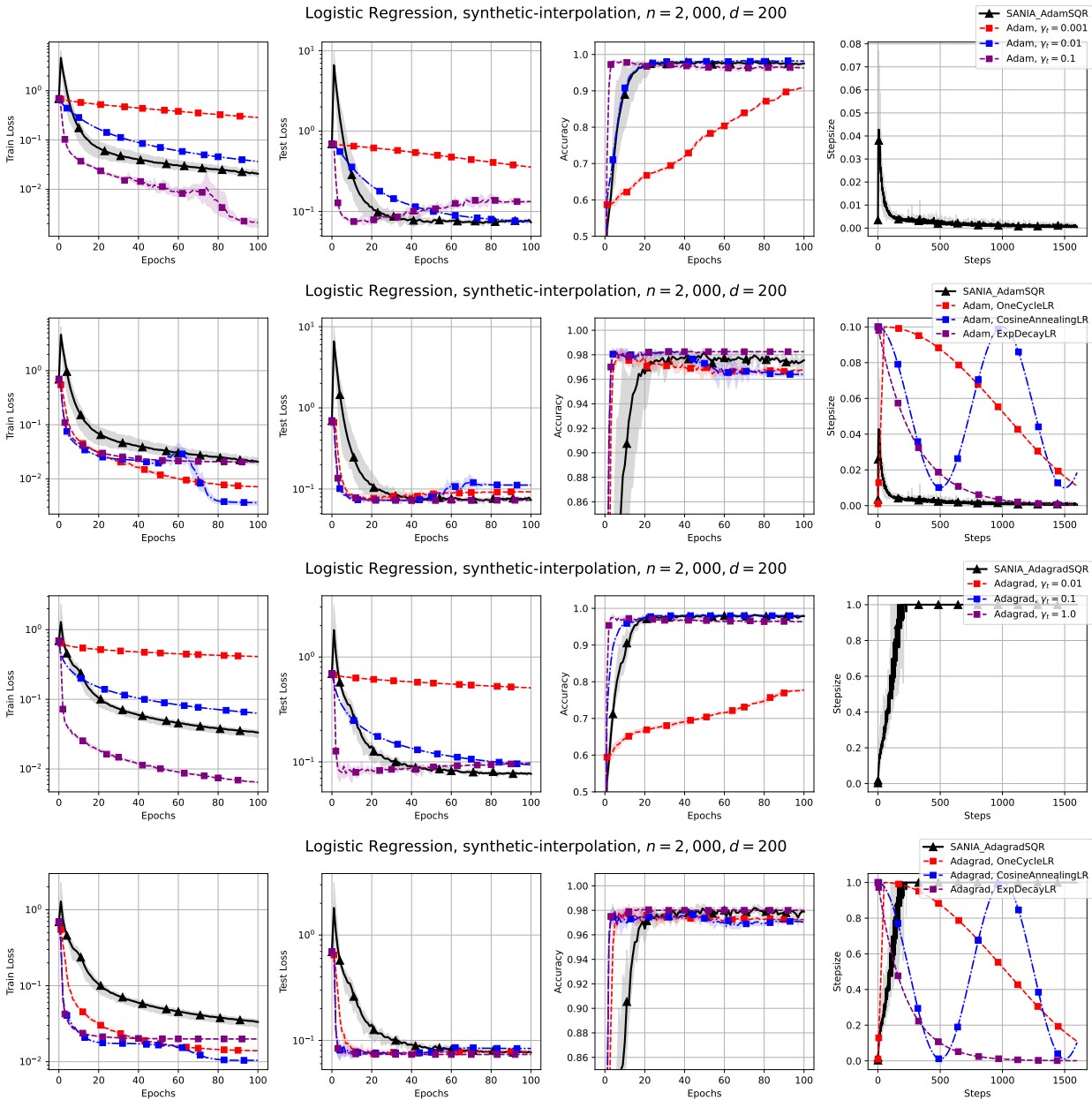

Figure 10: Evolution of metrics and step-sizes in SANIA, fine-tuned methods and learning rate schedules.

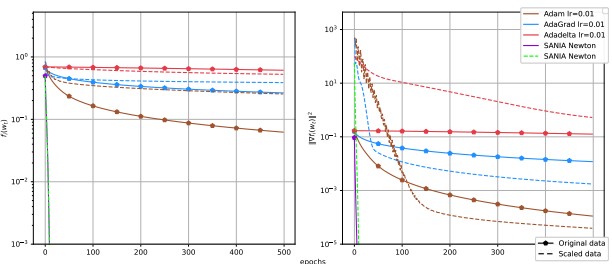

Figure 11: SANIA Newton compared to other adaptive methods on original and badly scaled ($k = 5$) synthetic binary classification dataset (batch size $= 100$) with logistic regression objective function.

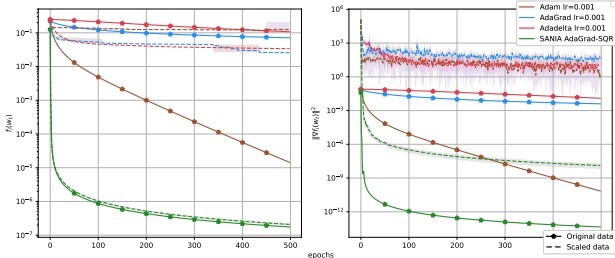

Figure 12: SANIA AdaGrad-SQR compared to other adaptive methods on original and badly scaled ($k = 10$) **mushrooms** dataset (batch size $= 256$) with non-linear least squares objective function.

Let us show the relation between them. For $v_t \leq 2$, SANIA step size is bigger but very close to PSPS, $2\lambda_t^{PSPS} \leq \lambda_t^{SANIA} \geq \lambda_t^{PSPS}$. However, for $v_t > 2$, the PSPS becomes more aggressive and $\lambda_t^{PSPS} > 1$, which is quite big for Newton-type methods and could be an issue when $f_i^*$ was chosen not accurate enough. We plot both of the step sizes to visualize the difference between them in Figure 14. Next, we provide the proofs for both step sizes inspired by proofs from (Schaipp et al., 2023).

---

**Lemma 12**

Let $f_i(x)$ be a convex function for all $i \in [1, \ldots, n]$ and have the same minimum $w^*$ (Assumption 1), $B_t \succ 0$ are positive definite matrices for $t \in [0, \ldots, T]$, $m_t = \nabla f_i(w_t)$, and $v_t = \frac{2(f_i(w_t) - f_i^*)}{\|\nabla f_i(w_t)\|_{B_t^{-1}}^2}$. Then for equation 58 method with the step size $\lambda_t \in (0, v_t)$, we have

$$\|w_{t+1} - w^*\|_{B_t}^2 < \|w_t - w^*\|_{B_t}^2. \tag{62}$$

Additionally, for $\lambda_t = \frac{f_i(w_t) - f_i^*}{\|\nabla f_i(w_t)\|_{B_t^{-1}}^2}$, we get

$$\|w_{t+1} - w^*\|_{B_t}^2 \leq \|w_t - w^*\|_{B_t}^2 - \frac{(f_i(w_t) - f_i^*)^2}{\|\nabla f_i(w_t)\|_{B_t^{-1}}^2}. \tag{63}$$

---

*Proof.* We start with the Polyak-step upper bound of the distance to the solution.

$$\|w_{t+1} - w^*\|_{B_t}^2 \stackrel{\text{equation 17}}{=} \|w_t - \gamma_t B_t^{-1} \nabla f_i(w_t) - w^*\|_{B_t}^2$$
$$= \|w_t - w^*\|_{B_t}^2 - 2\lambda_t \langle \nabla f_i(w_t), w_t - w^* \rangle + \lambda_t^2 \|\nabla f_i(w_t)\|_{B_t^{-1}}^2$$
$$\leq \|w_t - w^*\|_{B_t}^2 - 2\lambda_t (f_i(w_t) - f_i^*) + \lambda_t^2 \|\nabla f_i(w_t)\|_{B_t^{-1}}^2,$$

where in the last inequality we used the convexity of $f_i(x)$.

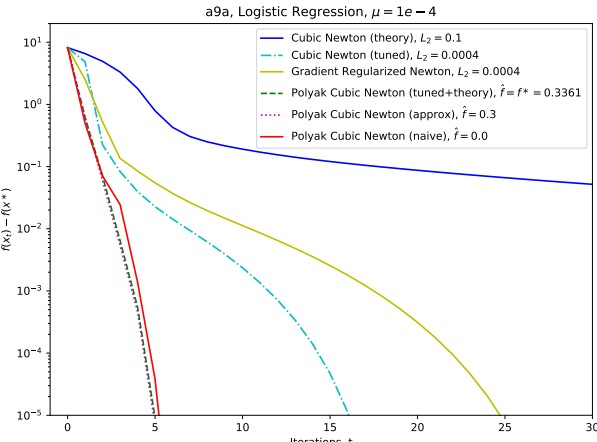

Figure 13: Gradient regularized(Cubic) Newton with Polyak step-size vs Cubic Newton methods for $\frac{\mu}{2}\|w\|^2$-regularized logistic regression

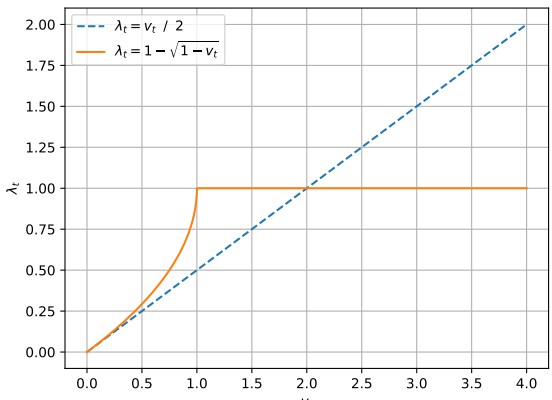

Figure 14: The comparison of step sizes $\lambda_t^{PSPS}$ (blue dashed line) from equation 60 and $\lambda_t^{SANIA}$ (orange dashed line) from equation 61.

For $\lambda_t \in (0, v_t)$ from equation 59, the right hand side is negative $-2\lambda_t \left(f_i(w_t) - f_i^*\right) + \lambda_t^2 \|\nabla f_i(w_t)\|_{B_t^{-1}}^2 < 0$, hence

$$\|w_{t+1} - w^*\|_{B_t}^2 < \|w_t - w^*\|_{B_t}^2$$

Next, if we optimize the right hand side by $\lambda_t$, we get the optimal $\lambda_t = \lambda_t^{PSPS} = \frac{v}{2}$ and

$$\|w_{t+1} - w^*\|_{B_t}^2 \leq \|w_t - w^*\|_{B_t}^2 - 2\lambda_t \left(f_i(w_t) - f_i^*\right) + \lambda_t^2 \|\nabla f_i(w_t)\|_{B_t^{-1}}^2$$
$$\leq \|w_t - w^*\|_{B_t}^2 - \frac{(f_i(w_t) - f_i^*)^2}{\|\nabla f_i(w_t)\|_{B_t^{-1}}^2}$$

$\square$

Next, we show the convergence theorem for the equation 58 method with the step size $\lambda_t = \frac{f_i(w_t) - f_i^*}{\|\nabla f_i(w_t)\|^2_{B_t^{-1}}}$.

Additionally, we assume that the preconditioning is not expanding $B_t \succeq B_{t+1} \succeq \nu$. It helps to work with the changing $B_t$-Euclidean norm. This assumption is satisfied for $B_t = I$ and for some Quasi-Newton updates.

---

**Theorem 2**

Let $f_i(x)$ be a convex $L_{\max}$-Lipschitz smooth function for all $i \in [1, \ldots, n]$ and have the same minimum $w^*$ (Assumption 1), $B_t \succ 0$ are positive definite matrices for $t \in [0, \ldots, T]$, $m_t = \nabla f_i(w_t)$, and $B_t \succeq B_{t+1} \succeq \nu$. Then for equation 58 method with the step size $\lambda_t = \frac{f_i(w_t) - f_i^*}{\|\nabla f_i(w_t)\|^2_{B_t^{-1}}}$, we get

$$\mathbb{E}[f(\hat{w}_T) - f^*] \leq \frac{2L_{\max}\|w_0 - w^*\|^2_{B_0}}{\nu T}, \tag{64}$$

where

$$\hat{w}_T = \frac{1}{T}\sum_{t=0}^{T-1} w_t \tag{65}$$

---

*Proof.* From equation 63 and the assumption that $B_t \succeq B_{t+1} \succeq \nu$, we get:

$$\|w_{t+1} - w^*\|^2_{B_{t+1}} \leq \|w_{t+1} - w^*\|^2_{B_t}$$
$$\overset{equation\ 63}{\leq} \|w_t - w^*\|^2_{B_t} - \frac{(f_i(w_t) - f_i^*)^2}{\|\nabla f_i(w_t)\|^2_{B_t^{-1}}}$$
$$\leq \|w_t - w^*\|^2_{B_t} - \frac{\nu(f_i(w_t) - f_i^*)^2}{\|\nabla f_i(w_t)\|^2} = \|w_t - w^*\|^2_{B_t} - \nu(f_i(w_t) - f_i^*)\frac{(f_i(w_t) - f_i^*)}{\|\nabla f_i(w_t)\|^2}$$
$$\leq \|w_t - w^*\|^2_{B_t} - \frac{\nu(f_i(w_t) - f_i^*)}{2L_{\max}},$$

where the last inequality is coming from the Lipschitz-smoothness of $f_i$: $\frac{1}{2L_{\max}} \leq \frac{(f_i(w_t) - f_i^*)}{\|\nabla f_i(w_t)\|^2}$.

Now, by taking the expectation and summing the previous inequality for $t = 0, \ldots, T - 1$, we get

$$\mathbb{E}[\|w_{t+1} - w^*\|^2_{B_T}] \leq \mathbb{E}[\|w_0 - w^*\|^2_{B_0}] - \sum_{t=0}^{T-1}\frac{\nu}{2L_{\max}}\mathbb{E}[(f_i(w_t) - f_i^*)].$$

Finally, by applying convexity to the average point $\hat{w}_T$, we get the convergence rate

$$\mathbb{E}[f(\hat{w}_T) - f^*] \leq \frac{1}{T}\sum_{t=0}^{T-1}\mathbb{E}[f(w_t) - f^*]$$
$$\leq \frac{2L_{\max}}{T\nu}\mathbb{E}\left[\|w_0 - w^*\|^2_{B_0} - \|w_T - w^*\|^2_{B_T}\right]$$
$$\leq \frac{2L_{\max}\|w_0 - w^*\|^2_{B_0}}{\nu T}.$$

$\square$

---

**Theorem 3**

Let $f_i(x)$ be a convex function for all $i \in [1, \ldots, n]$ and have the same minimum $w^*$ (Assumption 1), $B_t \succ 0$ are positive definite matrices for $t \in [0, \ldots, T]$, $m_t = \nabla f_i(w_t)$, $B_t \succeq B_{t+1} \succeq \nu$, and $\mathbb{E}[\|\nabla f_i(w_t)\|_{B_t^{-1}}^2] \leq G^2$. Then for equation 58 method with the step size $\lambda_t = \frac{f_i(w_t) - f_i^*}{\|\nabla f_i(w_t)\|_{B_t^{-1}}^2}$, we get

$$\min_{t=0,\ldots,T-1} \mathbb{E}[(f(w_t) - f^*)] \leq \frac{G\|w_0 - w^*\|_{B_0}}{\sqrt{T}}. \tag{66}$$

---

*Proof.* From equation 63 and the assumption that $B_t \succeq B_{t+1} \succeq \nu$, we get

$$\|w_{t+1} - w^*\|_{B_{t+1}}^2 \leq \|w_t - w^*\|_{B_t}^2 - \frac{(f_i(w_t) - f_i^*)^2}{\|\nabla f_i\|_{B_t^{-1}}^2} \tag{67}$$

$$\tag{68}$$

By taking the expectation on both sides, we get

$$\mathbb{E}[\|w_{t+1} - w^*\|_{B_{t+1}}^2] \leq \mathbb{E}[\|w_t - w^*\|_{B_t}^2] - \mathbb{E}\left[\frac{(f_i(w_t) - f_i^*)^2}{\|\nabla f_i\|_{B_t^{-1}}^2}\right]$$

$$\leq \mathbb{E}[\|w_t - w^*\|_{B_t}^2] - \frac{\mathbb{E}[(f_i(w_t) - f_i^*)^2]}{\mathbb{E}[\|\nabla f_i\|_{B_t^{-1}}^2]}$$

$$= \mathbb{E}[\|w_t - w^*\|_{B_t}^2] - \frac{(f(w_t) - f^*)^2}{\mathbb{E}[\|\nabla f_i\|_{B_t^{-1}}^2]}$$

$$\leq \mathbb{E}[\|w_t - w^*\|_{B_t}^2] - \frac{(f(w_t) - f^*)^2}{G^2}$$

We sum up and rearrange:

$$\frac{1}{T}\sum_{t=0}^{T-1} \mathbb{E}[(f(w_t) - f^*)^2] \leq G^2 \frac{1}{T}\sum_{t=0}^{T-1}\left(\mathbb{E}[\|w_t - w^*\|_{B_t}^2] - \|w_{t+1} - w^*\|_{B_{t+1}}^2]\right) \tag{69}$$

$$\leq \frac{G^2}{T}\left(\mathbb{E}[\|w_0 - w^*\|_{B_0}^2] - \underbrace{\mathbb{E}[\|w_T - w^*\|_{B_T}^2]}_{>0}\right) \tag{70}$$

$$\leq \frac{G^2}{T}\|w_0 - w^*\|_{B_0}^2 \tag{71}$$

$$\tag{72}$$

Due to Jensen's inequality $\mathbb{E}[\mathbf{X}^2] \geq \mathbb{E}[\mathbf{X}]^2$ and concavity of square root:

$$\mathbb{E}[(f(w_t) - f^*)^2] \geq \mathbb{E}[(f(w_t) - f^*)]^2 \tag{73}$$

$$\frac{1}{T}\sum_{t=0}^{T-1} \mathbb{E}[(f(w_t) - f^*)] \leq \sqrt{\frac{1}{T}\sum_{t=0}^{T-1} \mathbb{E}[(f(w_t) - f^*)]^2} \tag{74}$$

Using the above, we obtain:

$$\min_{t=0,\ldots,T-1} \mathbb{E}[(f(w_t) - f^*)] \leq \frac{1}{T}\sum_{t=0}^{T-1} \mathbb{E}[(f(w_t) - f^*)] \leq \frac{G\|w_0 - w^*\|_{B_0}}{\sqrt{T}}. \tag{75}$$

$$\square$$

**Remark 1.** The convergence proofs for the Gradient regularized Newton method with Polyak step-size equation 15 are presented in SP2 paper Li et al. (2023). Our main contribution in this part is deriving the explicit formula for general functions equation 16 and finding its connection to Cubic Regularized Newton.

**Remark 2.** The presented proofs do not cover all proposed methods and all step sizes. For example, from the current proofs $\lambda_t^{PSPS}$ is better than $\lambda_t^{SANIA}$ There are still open theoretical problems for us:
1) The convergence for expanding Euclidean norms, where $B_{t+1} \succeq B_t$.
2) Better convergence rates for Gradient regularized Newton method with Polyak step-size comparable to Cubic Newton convergence rates $O(T^{-2})$.
3) Better convergence rates for $\lambda_t^{SANIA}$ step-size in equation 58.
4) Extend the proofs to general $m_t$.

