# OpenReview forum: "SANIA: Polyak-type Optimization Framework Leads to Scale Invariant Stochastic Algorithms"
_TMLR — Rejected by TMLR_

### Review · Reviewer_SwjE · 2025-10-06

**Summary Of Contributions:**

This paper addresses the problem of step-size tuning for adaptive optimization methods. The paper presents an optimization framework that encompasses four known and existing optimization methods. The framework then leads to a new optimization method. Experimental evaluation covers few-layer neural networks on datasets such as MNIST, CIFAR-10, and SVHN.


Strengths:

* The paper introduction mentions the important problem of step-size tuning for adaptive optimization methods, which is a common problem in training Deep Neural Networks (DNNs).

* The mentioned framework generalizes over well-used methods like SGD, SPS, pSGD, AdaGrad and ADAM and this could lead to further exploration of these methods.

Weaknesses:

* As the contribution section opens "We present the General Framework for Preconditioned and Second-order Polyak methods". The "general framework" sounds interesting, but it is not clear what problem is being solved. Moreover, if there were any problem, it is not clear to what extent this framework covers some improvement in utility/efficiency/effectiveness or other advancement. Mention of the computational cost would be relevant.

* Most modern-day machine learning is done with transformer architectures for the deep neural network, but the experiment section does not mention these. I would argue that, precisely for training transformer models, step-size tuning is a crucial problem. Contrarily, for the few-layer neural nets that are used in the experiment section, step-size tuning is not an important problem, as a) step-size tuning can be done manually, and b) these architectures are usually used for tasks other than compute-intensive training tasks.

**Additional Comments:**

# Minor comments that are not part of the review

  * Introduction: Please use LaTex \citep{} for citations and only use \citet{} for in-line discussions of the citations.

  * 'To address problem equation 1, one of the fundamental techniques ...' -> 'To address problem Equation 1, one of the fundamental techniques employed ...'

  * 'Before delving into the details, we outline the primary contributions of this work:' This sentence reads like a sudden ending of the introduction. For any future version of the paper, I would encourage making the introduction a stand-alone introduction and discussing the method in the method section.

**Audience:**

No

**Audience Explanation:**

I will leave the matter of "audience match" to the AC. The paper focuses on optimization methods, which are of interest to some of the TMLR audience. However, some sections assume intimate prior familiarity with preconditioning and Polyak-type theory, which might not be applicable to the general TMLR audience (although I am sure that a short appendix section could go a long way). Pages 4, 5, 6, 7, 8, and 9 present a theoretical development of a framework and how four existing optimization methods fit in that framework. Such an exposition might find a better audience in a journal for the optimization community.

**Broader Impact Concerns:**

I could not find a Broader Impact Statement in the paper. However, I would say that Optimization methods and scale-invariant optimization do not need a Broader Impact Statement, and no ethical concerns are relevant.

**Claims And Evidence:**

Yes

**Claims Explanation:**

The evidence is accurate and clear for the chosen settings. However, I would argue that the scope is too narrow. Most modern-day machine learning is conducted using transformer architectures for deep neural networks, but the experiment section does not mention these. Moreover, the main contribution would be the alleviation of step-size tuning. Step-size tuning is relevant in a setting where, for example, training a large LLM takes weeks, and one cannot afford many separate runs to compare different step-sizes. In that case, an adaptive step-size method would be beneficial.

**Requested Changes:**

* Beyond guarantees and the effort of step-size tuning, many practitioners are interested in the computational cost of the optimization process. The paper mentions that SANIA is computationally efficient, but it would be helpful to have a few more details on the computational cost of the optimization process. Moreover, when exposing SGD, SPS, pSGD, AdaGrad, and ADAM as fitting in the framework, a comparison of the computational cost of these methods would be helpful.

* The experimental section would benefit from at least one transformer-like model and at least one large-scale dataset like GPT-2 or ImageNet-21k.

---

### Review · Reviewer_qCQ1 · 2025-10-19

**Summary Of Contributions:**

This paper presents SANIA, a constrained optimization framework (Eq. 6) that unifies Polyak-type methods, and proposes scale-invariant AdaGrad/Adam variants by removing square roots from preconditioning matrices. The framework recovers SGD, SPS, and preconditioned variants as special cases, with convergence analysis provided for PSPS under monotone decreasing preconditioning ($B_t \succeq B_{t+1} \succeq \nu I$). Experiments demonstrate the scale-invariance property across multiple datasets and show competitive performance with baseline methods on classification tasks.

**Additional Comments:**

The paper needs to address the major theoretical gaps, especially how it relates to their claims and the subsequent value of the paper to the community and TMLR audience.

**Audience:**

Yes

**Audience Explanation:**

The paper proposes a common framework that captures a range of existing popular optimization techniques, which could be a nice pedagogical resource.

**Claims And Evidence:**

No

**Claims Explanation:**

### Strength
1. Great introduction, I actually liked reading the intro which had a nice tone and progression of optimization techniques and placed their work among the existing literature.
2. Contains some good experimental analysis.

### Weaknesses
**Issues:**
1. The authors claim in their contribution: "*We introduce the new scale invariant versions of AdaGrad*... ", however they also mention in page 10: "Recently, scale invariant version of AdaGrad, named KATE, was proposed by Choudhury et al. (2024)." Both cannot be valid at the same time.
	1. On this note, could the authors also contrast their contributions from those of Choudhury et al (2024)?.
2. While comparing scale invariance, the paper doesn't compare with KATE (Choudhury et al., 2024) which is another scale-invariant AdaGrad variant.
3. Theorem 1 and 2 (in the appendix) seems to be the same theorems stated twice.

**No convergence analysis for the proposed SANIA step-size.**
The paper claims as a key contribution (Introduction, page 3): *"We introduce the new scale invariant versions of AdaGrad and Adam"* and presents the SANIA Quasi-Newton step-size $\lambda_t^{\text{SANIA}} = 1 - \sqrt{1-\upsilon_t}$ (Eq. 61) as the main methodological advance, used throughout SANIA AdaGrad-SQR, Adam-SQR, and Quasi-Newton (Algorithms 1-2, Eq. 17). However, Theorems 1-3 only prove convergence for the pre-existing PSPS baseline with $\lambda_t^{\text{PSPS}} = \upsilon_t/2$ (Eq. 60) from Abdukhakimov et al. (2023). The authors explicitly acknowledge in Remark 2 (page 40) that proving "better convergence rates for " remains an open problem. The claimed theoretical contribution thus analyzes an existing method rather than the proposed one. Without convergence guarantees or direct empirical comparison between $\lambda_t^{\text{SANIA}}$ vs $\lambda_t^{\text{PSPS}}$, there is no rigorous justification for why practitioners should prefer the proposed step-size over the simpler, theoretically-grounded baseline. Can the authors comment on this?

**Theoretical analysis relies on potentially contradictory assumptions**
Theorems 1-3 require $B_t \succeq B_{t+1} \succeq \nu I$ (monotone decreasing preconditioning), but all practically relevant preconditioners violate this. Standard AdaGrad has $B_t = \text{diag}(\sqrt{\sum_{j=1}^t g_j^2})$ which increases ($B_{t+1} \succeq B_t$), directly contradicting the assumption. Adam fluctuates (typically increasing early), and Hutchinson fluctuates arbitrarily. **Critically, the paper's own SANIA AdaGrad-SQR ($B_t = \text{diag}(\sum_{j=1}^t g_j^2)$, Eq. 19) and Adam-SQR (Eq. 20) also increase or fluctuate, placing them outside Theorems 1-3's scope.** The theory essentially only covers $B_t = I$ (vanilla SPS), already analyzed in Loizou et al. (2021). Combined with the issue of step size analysis, the paper's main contributions: SANIA AdaGrad-SQR and Adam-SQR using $\lambda_t^{\text{SANIA}}$, have no convergence guarantees for either their step-size or preconditioning. The theory analyzes PSPS under impractical assumptions while experiments use different, unanalyzed methods. Can the authors comment on this?

**Justification of $\lambda_t = 1$ fallback when $\upsilon_t > 1$**
The $\lambda_t = 1$ fallback when $\upsilon_t > 1$ (Eq. 18) is theoretically unjustified. When the constrained optimization problem in Lemma 8 becomes infeasible, the paper defines $\lambda_t = 1$ as a "minimum of the constraint" without formal derivation. This heuristic fallback is not analyzed in Theorems 1-3, and its impact on convergence remains unproven. Can the authors provide justification for this choice?

**Requested Changes:**

**Questions**
- - Can the authors provide convergence rates for $\lambda_t^{\text{SANIA}}$ (Eq. 61) comparable to the rates proven for $\lambda_t^{\text{PSPS}}$ in Theorems 1-3?
- Why is $\lambda_t^{\text{SANIA}}$ preferred over simpler $\lambda_t^{\text{PSPS}} = \upsilon_t/2$?
- How do results extend when $B_t \succeq B_{t+1}$ doesn't hold (standard AdaGrad/Adam)?
- What is the computational overhead of cubic Newton variant (Eq. 16)?

---

### Review · Reviewer_zNSm · 2025-10-28

**Summary Of Contributions:**

This work introduces a unified optimization framework called SANIA (Scale-Invariant Adaptive Polyak Framework) that generalizes several adaptive and Polyak step-size methods. Within this framework, the authors propose:
1. A Stochastic Cubic Newton method with Polyak step-size;
2. Two scale-invariant adaptive optimizers, namely SANIA AdaGrad-SQR and SANIA Adam-SQR, derived by removing the square-root operation from classical preconditioners;
3. A theoretical argument that some of SANIA methods are affine or scale invariant;
4. Experimental evaluation of SANIA AdaGrad-SQR and Adam-SQR on standard vision datasets, showing similar or slightly better results than Adam, AdaGrad, and KATE.

**Additional Comments:**

The experimental results of the proposed SANIA variants seems do not show a clear or significant improvement over traditional optimization methods such as Adam and AdaGrad. Moreover, the appendix section on experiments provides extended numerical results but offers no detailed discussion or interpretation of those outcomes, making it difficult to understand under what conditions the proposed methods truly outperform existing approaches.

**Audience:**

Yes

**Audience Explanation:**

Yes.
1. The SANIA framework covers some classical optimization methods such as SGD, ADAM and their variants and inspires the proposal of new algorithms.
2. Five new optimization algorithms with Polyak step-size have been proposed, which have desirable properties e.g. affine and scale invariance and achieve comparable performance in the experiments under both convex and non-convex settings.

**Broader Impact Concerns:**

No broader impact concerns are identified.

**Claims And Evidence:**

Yes

**Claims Explanation:**

While the general framework is mathematically consistent and clearly explained, some claims are not accurate and convincing enough.
1. Since the PSPS is not original, more explanations are needed to illustrate the difference between Theorem 1 and the existing convergence analyses of PSPS, or state that Theorem 1 is the first convergence results of PSPS.
2. The “first stochastic cubic Newton with Polyak step-size” claim lacks convergence analysis. The author mention that the convergence analysis has been given in SP2 paper Li et al. (2023). However, more explanations (the difference with SP2) and detailed proofs should be given. Also, it's better to include the experiments of this method in the main text.
3. Comparisons between SANIA AdaGrad-SQR and KATE are incomplete. These two methods both remove the square root from AdaGrad update. What's the difference?
4. There is a lack of theoretical demonstrations of the advantages of the proposed methods, that is, whether these methods achieve the state-of-the-art convergence rate.

**Requested Changes:**

1. Remark or explanation on Theorem 1 and Lemma 1 (at least including the novelty, comparison and whether the assumptions on $B_t$ and $\lambda_t$ are reasonable?)
2. Give some intuitions and explanations in Section 2.1 to support the statement mentioned in contribution: the SANIA provides valuable insights into Polyak step-size methods.
3. The convergence analyses of all the newly proposed methods should be given. Comparisons with the existing theoretical works are also needed.

---

### Decision · Action_Editor_ovre · 2025-12-01

**Recommendation:** Reject

**Audience:**

Yes

**Audience Explanation:**

The paper proposes a general framework which covers many adaptive stochastic optimization algorithms. This framework should be interesting to the TMLR audience since stochastic optimization are popular optimization methods to train deep neural networks and transformers.

**Claims And Evidence:**

No

**Claims Explanation:**

The paper presents a general framework which includes Polyak-type methods. Within this framework, the paper further proposes scale-invariant AdaGrad/Adam variants by removing square roots from preconditioning matrices. A nice property is that it does not require the manual fine-tuning of the step-size.

Meanwhile, reviewers also mention that there are issues in the theoretical analysis. For example, the theoretical analysis imposes some assumptions which are contrary to the practical cases. In particular, the analysis requires monotone decreasing preconditioning, while in practice we often use monotone increasing preconditioning. Therefore, the theoretical analysis is not quite convincing. Other reviewers also proposed some suggestions on the experimental analysis.

The authors did not post a revision and the responses to the review comments, and a revision.  Therefore, the issues indicated by the reviewers are not addressed.

**Resubmission Of Major Revision:**

The authors may consider submitting a major revision at a later time.